# Collagen Matrix to Restore the Tympanic Membrane: Developing a Novel Platform to Treat Perforations

**DOI:** 10.3390/polym16020248

**Published:** 2024-01-15

**Authors:** Mikhail Svistushkin, Svetlana Kotova, Anna Zolotova, Alexey Fayzullin, Artem Antoshin, Natalia Serejnikova, Anatoly Shekhter, Sergei Voloshin, Aliia Giliazova, Elena Istranova, Galina Nikiforova, Arina Khlytina, Elena Shevchik, Anna Nikiforova, Liliya Selezneva, Anastasia Shpichka, Peter S. Timashev

**Affiliations:** 1Department for ENT Diseases, Sechenov First Moscow State Medical University (Sechenov University), 8-2 Trubetskaya St., Moscow 119991, Russia; svistushkin_m_v@staff.sechenov.ru (M.S.); zolotova_a_v@staff.sechenov.ru (A.Z.); nikiforova_g_n@staff.sechenov.ru (G.N.); khlytina_a_a@staff.sechenov.ru (A.K.); shevchik_e_a@staff.sechenov.ru (E.S.); james.ann@mail.ru (A.N.); selezneva_l_v@staff.sechenov.ru (L.S.); 2Institute for Regenerative Medicine, Sechenov First Moscow State Medical University (Sechenov University), 8-2 Trubetskaya St., Moscow 119991, Russia; kotova_s_l@staff.sechenov.ru (S.K.); fayzullin_a_l@staff.sechenov.ru (A.F.); antoshin_a_a@staff.sechenov.ru (A.A.); serezhnikova_n_b@staff.sechenov.ru (N.S.); shekhter_a_b@staff.sechenov.ru (A.S.); voloshin_s_yu@staff.sechenov.ru (S.V.); gilyazova_a_n@staff.sechenov.ru (A.G.); istranova_e_v@staff.sechenov.ru (E.I.); timashev_p_s@staff.sechenov.ru (P.S.T.)

**Keywords:** tympanic membrane, perforation, collagen, electrophoretic deposition, SBA-EPD

## Abstract

Modern otology faces challenges in treating tympanic membrane (TM) perforations. Instead of surgical intervention, alternative treatments using biomaterials are emerging. Recently, we developed a robust collagen membrane using semipermeable barrier-assisted electrophoretic deposition (SBA-EPD). In this study, a collagen graft shaped like a sponge through SBA-EPD was used to treat acute and chronic TM perforations in a chinchilla model. A total of 24 ears from 12 adult male chinchillas were used in the study. They were organized into four groups. The first two groups had acute TM perforations and the last two had chronic TM perforations. We used the first and third groups as controls, meaning they did not receive the implant treatment. The second and fourth groups, however, were treated with the collagen graft implant. Otoscopic assessments were conducted on days 14 and 35, with histological evaluations and TM vibrational studies performed on day 35. The groups treated with the collagen graft showed fewer inflammatory changes, improved structural recovery, and nearly normal TM vibrational properties compared to the controls. The porous collagen scaffold successfully enhanced TM regeneration, showing high biocompatibility and biodegradation potential. These findings could pave the way for clinical trials and present a new approach for treating TM perforations.

## 1. Introduction

The closure of tympanic membrane (TM) perforations is one of important problems in modern otology. Perforation is considered persistent if it has not closed on its own in three months [1]. The presence of a persistent TM defect leads to the development of chronic inflammation in the middle ear, which may be accompanied by a discharge from the ear, conductive hearing loss, high risk of inner ear damage, and regional and intracranial complications [2,3,4,5].

Transforming a sound wave into a mechanical vibration, the TM plays a key role in sound conduction, since it represents a distinct border between the middle ear and inner ear [6]. The dynamic properties of the TM are anatomically determined by the mobility of the chain of the auditory ossicles, as well as by the state of the TM itself, its thickness and structure [6]. Correct TM functioning directly depends on its vibrational competence which is determined by the physical properties of its middle fibrous layer [7]. In turn, the middle layer is represented by two layers of collagen fibers, radially and circularly oriented [8]. These collagen fibers are responsible for the viscoelastic properties of the TM, which, in turn, characterize the TM mechanics [5,9]. Perforation of the TM leads to changes in its mechanical properties and hearing impairment. Therefore, in the field of otolaryngology, the search for materials with structures and mechanical properties similar to those of the TM, and their use as scaffolds to improve the regeneration of defects and restore the vibrational competence of the TM, remain a crucial issue.

To restore sound conduction and prevent the development of otitis media and its complications, any defect of the TM must be closed. The closure of a TM perforation is aimed at the restoration of not only its integrity but also its mechanical and acoustical properties playing a crucial role in sound transmission [10,11]. A multitude of techniques for the closure of TM perforations is described in the literature, with the application of various autografts (fascial flap, perichondrium, fragments of cartilage, multilayered flaps) and other materials [5,9,12,13,14,15]. The efficiency of the surgical treatment depends on many factors, including the applied transplant’s thickness, the surgeon’s experience, and the preparation of the defect edges prior to the procedure [16,17]. However, in any case, there is still a risk of graft failure, its incomplete engraftment or rejection in the post-surgical period, i.e., of TM re-perforation. Furthermore, optimal functional results are not achieved in all cases [12,13,15,18].

The development of alternative ways to restore TM defects, enabling the avoidance of problems inherent in surgical treatment, is an important problem in modern medicine. Significant attention has been paid lately to the methods of re-generative medicine in the closure of TM perforations [18]. Since the 1990s, with the development of tissue engineering, synthetic materials have demonstrated great potential for TM restoration [19]. The use of tissue-engineered scaffolds provides significant advantages such as the reduction in surgery time and minimizing surgical tissue damage, reduction in economic expenses, and an increase in the percentage of positive morphofunctional results [5,14].

In spite of the TM’s inherent regenerative capacity, a support under the growing epithelial layer is required for its restoration [13].

The published research includes the use of scaffolds, regulatory factors, as well as cell technologies both separately and in various combinations [20,21,22,23]. The application of non-autologous materials in the closure of TM perforations provides essentially reduced invasiveness of the procedure and the possibility of transplant placement in the outpatient setting [23,24]. Different scaffolds or transplants are applied in TM bioengineering, which serve as supports to facilitate the migration of cells and nutrients towards the perforation [18]. As opposed to conventional transplants retained in vivo, a bioengineered construct is meant to degrade with time and to be replaced with new tissue [15]. We view the perfect purpose of tissue engineering techniques as the restoration of the three-layered TM structure which is crucial for sustaining its mechano-acoustic properties. Various scaffolds, stem cells and growth factors are used for the TM regeneration [13]. The above materials have been applied both separately and in various combinations, according to the data of different studies [13,15,18]. All these components play a vital role both for cell differentiation, proliferation and migration and for the creation of a proper biochemical microenvironment to prepare and stimulate tissue healing [5,20,21].

To date, many scaffolds have been tested in the closure of both acute and chronic TM perforations [4,5]. Collagen-based materials are used frequently for this purpose, with some differences in their physicochemical properties [25,26]. The application of collagen or gelatin sponges is most popular in clinical practice, including their combination with different growth factors [22,27,28,29,30]. Collagen has a unique biological activity, collagen-based sponges have demonstrated effective absorption of the wound exudate and provide a humid environment and the prevention of infection and mechanical injury when in contact with a wound. It should be noted that this material’s porosity enables the transport of cells and nutrients [31]. Thus, collagen-based biomaterials are of primary interest for the restoration of TM defects due to their high biocompatibility and versatility.

Recently, we created a robust collagen membrane with superior mechanical characteristics, using semipermeable barrier-assisted electrophoretic deposition (SBA-EPD) [32]. The key feature of this technique, as compared to conventional EPD, is the application of a semipermeable barrier of regenerated cellulose installed into the electrochemical cell, which divides the cathode and anode spaces. The anode space contains the collagen suspension, while the cathode space contains only water. The EPD is performed in a multicyclic layer-by-layer manner; the resulting collagen membrane is peeled off the surface of the cellulose barrier. The developed technique has been shown to produce defect-free uniform membranes with no toxicity revealed in the in vitro and in vivo studies, and with a good biodegradation profile [32].

Here, we applied a collagen graft in a porous (sponge) form that was prepared from the SBA-EPD membranes by post-treatment procedures, in the surgical treatment of acute and chronic TM perforations in a chinchilla model. The TM regeneration after the graft implantation was studied, along with the morphological analysis of the restored TM, and the potential of the developed technique’s application in clinical practice was estimated.

## 2. Materials and Methods

### 2.1. Creation of the Collagen Matrix

#### 2.1.1. Preparation of a Collagen Suspension

Freshly frozen bovine tendons were thawed, purified from excessive tissues, and cut into fragments with the thickness of 1 cm. The tendon fragments underwent four 12 h long treatments with 0.5 M NaCl, followed by homogenization in 0.8 M acetic acid (the pH of suspension was 3.24). Pepsin (1 mg/mL) was added to the resulting suspension; the hydrolysis proceeded for 2 days. Then, the pH was increased to 7.5 using 1 M NaOH to stop the hydrolysis, and the collagen suspension was precipitated with a 120 mg/mL NaCl solution. The deposited collagen was re-dissolved in 0.8 M acetic acid (the pH of suspension was 2.7) and dialyzed against 0.5 M acetic acid for 3 days, the dialysis solution being replaced every day. After dialysis, the pH of the suspension was 2.9. After each dialysis procedure, we measured the pH of the resulting solution. The reproducibility of our results is guaranteed by the performed pH measurements, and after each dialysis, the pH of the solutions was stably equal to ~2.9. Therefore, we concluded that the performed dialysis procedure was well standardized. The final collagen concentration suspension was determined by gravimetry.

#### 2.1.2. Preparation of a Collagen Membrane by Semipermeable Barrier-Assisted Electrophoretic Deposition

A collagen matrix was prepared by electrodeposition as previously described [1]. A collagen suspension of 5 mg/mL was used. The deposition was conducted in an electrochemical cell, divided by a semipermeable barrier of regenerated cellulose (Sigma-Aldrich, St. Louis, MO, USA) into two parts, each of which contained either the cathode or anode. The collagen suspension was poured into the anode part of the cell, while distilled water was poured into the cathode part of the cell. The cathode and anode were plate electrodes, connected to a DC source with a voltage of 60 V. The SBA-EPD-obtained membranes were carefully detached from the semipermeable barrier and treated with isopropanol for 20 min, then they were dried in a laminar flow hood.

#### 2.1.3. Post-Treatment of the Collagen Membrane

Creation of porous and perforated collagen forms was achieved via controlled chemical crosslinking of SBA-EPD-produced collagen membranes (in PBS) with 0.05% glutaraldehyde in PBS (Sigma-Aldrich, St. Louis, MO, USA) for 30 min; then they were placed into fresh PBS and mechanically perforated on both sides using a cosmetic mesoroller with titanium needles, washed with PBS, and lyophilized at −40 °C for 24 h. As a result, the pH of their extracts was around 6.5–7.5.

### 2.2. Collagen Matrix Characterization

Visualization of the collagen matrix microstructure was performed by using an EVO LS10 scanning electron microscope (SEM, Zeiss, Germany) in the low vacuum mode (EP, 70 Pa), at the accelerating voltage of 21 kV, and a current on the sample of 40–70 pA, using a back-scattered (reflected) electron detector.

A LEICA DM4000 B LED microscope (Leica Microsystems, Wetzlar, Germany) was used for the examination of collagen matrix cross section. The specimens were stained by hematoxylin and eosin and studied using phase-contrast microscopy.

The swelling test was performed in accordance with the protocols described elsewhere [33,34,35]. The dried samples of collagen matrix (*n* = 5 per group) were first weighed, then they were placed in phosphate buffered saline (PBS; pH 7.5; D8537; Sigma-Aldrich, St. Louis, MO, USA) at 4 °C for 8 h. Subsequently, the excess moisture was removed by filter paper, and the samples were weighed again. The swelling ratio was calculated as a change (in %) in the mass of the wet and dry samples according to the formula:$$S=\frac{m_{w}-m_{d}}{m_{d}}100\%$$
where *S*—swelling ratio, *m_w_*—mass of the wet sample, *m_d_*—mass of the dry sample.

The measurement of collagen matrix shrinkage temperature (*n* = 5 per group) was performed by the hydrothermal method using a laboratory-made device. For this, 20 × 3 mm strips were cut and placed in a glass tube that was immersed in a water bath with distilled water. The water temperature was gradually increased by about 5 °C/min. The temperature at which the shrinkage was observed was recorded by thermometer.

The mechanical parameters of collagen matrices (*n* = 5 per group) were tested in the wet conditions by Mach-1 v500csst micromechanical testing system (Biomomentum Inc., Laval, QC, Canada). The strain at failure, and the modulus of elasticity (Young’s modulus, calculated in the linear region of a stress–strain curve) were measured in the uniaxial tension mode for at least five 30 × 5 mm^2^ rectangular specimens. Uniaxial tension was performed at a rate of 0.1 mm/s until failure was achieved. The parameters were measured from the deformation curves according to the manufacturer’s protocol.

### 2.3. Tympanic Membrane Regeneration Study

#### 2.3.1. Study Design

The animal experiments were approved by the local Ethical Committee at the Sechenov University (Protocol No. 11–23, 15 June 2023). In this study, 12 male chinchillas were used, 6 months of age and 500–700 g in weight, from the central animal facility of the Sechenov University. The conditions of animal housing and feeding met the requirements of the Russian “Guide for the housing and care of laboratory animals. Regulations on the housing and care of laboratory animals and rabbits, as of 1 July 2016” (GOST 33216-2014) and of the European Convention for the Protection of Vertebrate Animals used for Experimental and Other Scientific Purposes [36,37].

By its design, it was a controlled experimental study (Figure 1). The experiments were divided into two blocks based on the model of the TM perforation: (1) acute; (2) chronic.

In the first block (acute perforations), a bilateral TM defect was created surgically in 6 animals. Right-side perforations were left for observation only (control Group 1, *n* = 6). Left-side perforations were closed using a porous collagen graft immediately after their creation (experimental Group 2, *n* = 6).

In the second block (chronic perforations), at the first stage a persistent TM perforation was created by folding the edge of a perforation followed by placing a tympanostomy tube for 30 days, according to our previously developed technique [12]. After tube removal, dynamic observation of the formed persistent TM perforation with a callous edge was performed using otovideoendoscopy for 21 days. Then, the chronic perforation edge was de-epidermized. On the right side, the closure of perforation was not performed (control Group 3, *n* = 6). On the left side, the perforations were closed with a porous collagen graft (experimental Group 4, *n* = 6). For the comparative evaluation of the morphological properties, 6 preparations of intact TMs of chinchillas from the Biobank of the Sechenov University were used.

#### 2.3.2. Surgical Procedures

The surgical interventions were performed under the conditions of the operating room of the animal facility. All the manipulations were performed under drug-driven analgosedation using a tiletamine-zolazepam solution (Zoletil 100), based on the 10–15 mg/kg dose, and a xylazine solution (Rometar) based on the 1–2 mg/kg dose. A Karl Storz endoscope with d = 2.7 mm, l = 10 cm, 300 (Karl Storz, Berlin, Germany) was used for intraoperative visualization. In the postsurgical period, dynamic observation of the operated animals was conducted; no drug therapy was applied.

Creation of a defect model of acute TM perforation: Acute TM perforation was inflicted in 6 chinchillas on both right and left side under a visual endoscopic control (first block of the study, acute perforations). Using a microdissection needle, the TM was punctured in its central part, and the edges of the tear were dissected up to the diameter of 3 mm. All the right-side perforations (*n* = 6) formed Group 1 (control), and no further manipulations were performed on them. All the left-side perforations were closed with a porous collagen graft immediately after defect creation (experimental Group 2).

The technique for the closure of an acute perforation with a porous collagen graft: The sponge form of the collagen matrix was soaked in saline for 15 min. Then, an implant with the diameter of 4 mm was cut. The implant was directly placed on the TM surface in the perforated region with the use of microforceps and a microdissection needle so that the perforation’s edges were overlayed by 1 mm (Figure 1).

Creation of a model defect of chronic TM perforation: The experimental model of chronic TM perforation was applied to 6 chinchillas of the second block of the study (chronic perforations) on both right and left side. At the first stage, a myringotomy hole was formed in the rear-lower quadrant using a microneedle from the standard otiatric surgical set. Then, using the same instrument, four radial cuts at 12, 3, 6, and 9 o’clock positions were performed, and the formed flaps were folded inside, towards the medial surface of the TM. Further, after folding the edge of the perforation, a tympanostomy tube was inserted into the defect for 30 days. After 30 days, the tube was removed, and the formed TM perforation underwent dynamic observation for 21 days in order to control a spontaneous closure (Figure 1).

At the second stage, in the group of right-sided chronic perforations (*n* = 6), a dissection of the epidermal edge was performed (“refreshing the edges”); the perforation was not closed, this group was control Group 3. The left-side defects (*n* = 6) after “refreshing the edges” were closed with a graft of the porous collagen using the same technique as the one used in the closure of acute perforations, and included in experimental Group 4 (Figure 1). Refreshing the edges of the chronic perforation is a surgical dissection of a thin tissue strip at the perforation edges. The effectiveness criterion at this tympanoplasty stage is the appearance of fresh blood on the edge surface that is essential for a good adhesion of an implant. In the case of the acute perforation, the small amount of blood appears due to the initial surgical trauma.

Postsurgical follow-up. In the postsurgical period, the dynamic observation of the operated animals was conducted; no drug therapy was applied.

### 2.4. Endo-Otoscopy

The endoscopic control of the regeneration with photoregistration was performed in all the animals: in Groups 1 and 2—on days 14 and 35 after the creation of a perforation; in Groups 3 and 4—on days 14 and 35 after the de-epidermization of the persistent perforation’s edge. For a comparative estimation of Groups 1 and 2 on day 14, the presence or absence of the signs of the TM and middle ear inflammation (hyperemia, edema, injection of the mucosa’s blood vessels, mucous-purulent exudate, formation of granulations, unchanged size of the perforation) were registered. The presence or absence of these signs were assigned qualitative values of “1” or “0”, respectively. On day 35, the assessment of the perforation closure was performed in all the animals, and in Groups 1 and 2 of acute perforations, the degree of scar development was additionally registered as the following status: «0»—absence of a visually discernible scar, and «1»—the visual presence of scar tissue. To estimate the statistical significance of the groups by these signs, Fisher’s exact test was used, with the threshold value of *p* = 0.1. The number of independent assessments of each endoscopic photo was three; scores were provided blindly by three otolaryngologists.

### 2.5. Morphology Study

The animals of the control and experimental groups were removed from the experiment on day 35 after the creation of an acute perforation (Groups 1 and 2) and on day 35 after the de-epidermization of the edge of a persistent perforation (Groups 3 and 4) by an intramuscular injection of the tiletamine-zolazepam solution. Under the conditions of the animal facility, a block-resection of the temporal bone with a fibrous ring and TM inside was performed, to leave the TM as a whole, and the preparation was sent for the histological study and evaluation of the vibrational properties.

Tissues fixed in 10% neutral buffered formalin (7.0–7.4 pH; sodium phosphate monobasic 4.0 g, sodium phosphate dibasic 6.5, formaldehyde 37% 100 mL, distilled water ad 1000 mL; B06-001/L, HistoSafe original, Biovitrum, Russia) were then decalcified, dehydrated and embedded in paraffin blocks. Sections 4 µm thick were stained with hematoxylin-eosin and with Mallory’s trichrome stain and studied with a Leica DM 4000 B LED universal microscope equipped with a Leica DFC 7000 T camera under the control of the LAS V4.8 software (Leica Microsystems, Wetzlar, Germany). Samples were studied with bright-field and phase-contrast optical microscopy. Low-magnification images of histological slides were obtained with a Bresser USB microscope (Bresser, Rhede, Germany).

### 2.6. Morphometric Analysis

Morphometric analysis of the histological samples was performed by two blinded pathologists. In each sample, in the course of the collagen membrane’s biological integrtion in the TM perforations, the signs of inflammation and regeneration (TM thickness, density of collagen fibers, inflammatory infiltration, vascularization) were estimated on a 5-point scale (0—no, 4—highest intensity) (Table A1, Table A2, Table A3 and Table A4).

### 2.7. Study of Vibrational Properties and the Amplitude-Frequency Characteristic of the Tympanic Membrane

The study of vibrational properties and the amplitude-frequency characteristic (AFC) of the TM was conducted using an experimental setup with a fiber optic probe. The experimental setup consists of a macrosample of the TM installed in a silicone conical holder; a fiber optic probe for the detection of vibrational signals; an optoelectronic laser block with an accessory red laser to visualize the spot of the laser probing; and a two-channel sound card with an analog-to-digital converter (ADC) to record signals in real time to a personal computer (Figure A2). The recorded signal is processed using special software with the Fast Fourier Transformation (FFT) function for indication of instant AFCs during recording or at a certain point in the record afterwards. The setup is based on a highly sensitive vibrometer with a laser fiber optic probe at the end of a single mode fiber with a ceramic FC connector with a flat face. The sensitive element of the setup is a low Q factor Fabry–Perot interferometer formed by the flat face of the fiber optic tip with 4% Fresnel reflection and the first boundary of the vibrating surface of the TM. The measurements were performed immediately after the autopsy, the TM macrosample represented the whole TM structure with the diameter of ~8 ± 1 mm in the bony ring of the external auditory canal and bony walls of the tympanic cavity with the intact chain of the auditory ossicles. The measurements of the vibrational characteristics were performed from the inner side of the TM, the stimulating audio signal of 40–60 dB was applied to the outer side of the TM. Monochromatic tone sounds at the frequencies ranging from ~20 Hz to 10–20 kHz, as well as combinations of up to 5 frequencies simultaneously, were used as test audio signals to measure the sensitivity by the presence of vibrational responses. To acquire the integral AFC of the studied sample, a linear frequency sweep from the low to the ultimately high (~10–20 kHz) frequencies was used that provided the envelope representing the required AFC of the TM. The acquisition of both the individual responses at selected frequencies of the audio stimulation using FFT and the integral envelope function of the mentioned responses during the linear stimulating frequency sweep was performed in 10 s. Using this technique, we measured three intact TM and experimental ones: Group 1—acute perforation, control, *n* = 3 (spontaneous closure in 35 days); Group 2—acute perforation, porous collagen membrane, *n* = 3 (closure in 35 days); and Group 4—chronic perforation, porous collagen membrane, *n* = 3 (closure in 35 days). In Group 3, there were no cases of spontaneous closure of the TM perforation; therefore, it was not possible to measure the vibrational properties of the perforated TM. The number of independent measurements was five.

### 2.8. Statistical Analysis

The statistical analysis of the experimental data was performed with the use of the GraphPad Prism 8.00 software (GraphPad Software, Boston, MA, USA). The differences in the macroscopic and histological scores were estimated using the Kruskal–Wallis test with the Dunn’s test for multiple comparisons. *p* ≤ 0.05 were considered as significant. The results of the statistical analysis are presented as histograms of median values and a 95% confidence interval (CI).

## 3. Results

### 3.1. Collagen Matrix Characteristics

The prepared collagen matrices (Figure A1) had perforations on both sides, while the cross sections were lamellar and porous (Figure 2). The pores within the collagen matrix body were not interconnected and resembled a honeycomb (Figure A3); they were formed between the layers of the electrophoretically deposited collagen.

The thickness and swelling values of the porous collagen membranes differed from those of the initial SBA-EPD matrices (Table 1); however, their shrinkage temperatures were identical (*n* = 5 measurements were performed for each parameter). As expected, the porous collagen membranes were thicker and had more swelling than the initial SBA-EPD membranes. The shrinkage temperature of both forms of collagen matrix was the same and equaled 54–56 °C.

When testing the mechanical properties, as shown in Table 2, the Young’s modulus and strain at failure of the porous collagen form decreased compared to the initial SBA-EPD matrices by 12 MPa and 14%, respectively, (*n* = 5 measurements were performed for each parameter).

### 3.2. Endo-Otoscopy

Endo-otoscopy of the first block of the experimental study (acute perforations). The endoscopic control performed 14 days after the creation of acute perforation showed that, in control Group 1, in 5 animals (83.3%), inflammatory phenomena were observed such as the formation of granulation tissue, hyperemia and edema of the TM around the perforation. No complete closure was observed in any animal (Figure 3A). In one animal (16.7%), TM thickening was noted, the perforation was covered with a crust, a weak tendency to perforation narrowing was noted. In the 2d experimental group (collagen), after 14 days, the endoscopic examination revealed the collagen graft applied to the area of perforation. In five animals (83.3%), the graft tightly adjoined the intact TM tissues around the perforation, no reactive phenomena were present, the perforation size could not be determined without biomaterial removal. Some resorption and thinning of the graft should be mentioned; however, its macrostructural integrity was preserved (Figure 3A). In one animal (16.7%), the collagen graft did not cover the perforation edges, it shifted towards the TM periphery; reactive phenomena and TM edema around the perforation were also noted. In this last case, the perforation edges were refreshed, and the collagen graft was applied again. When comparing Groups 1 and 2 for the presence of inflammatory changes, the differences were significant (Fisher’s exact test, two-sided *p*-value = 0.0801), being less pronounced in Group 2 with the application of a collagen graft (Figure 3B).

In control Group 1 and experimental Group 2, after 35 days, the closure of the TM perforation occurred in all the observation objects (six TMs in each group). The endoscopic examination in control Group 1 revealed the presence of a rough scar at the site of a TM perforation in four animals (66.7%) (Figure 3A). In two animals (33.3%), the scarring was less pronounced and represented a thickened scar tissue bundle. In experimental Group 2, which used porous collagen, insignificant residual fragments of an unresorbed collagen graft were found on the TM surface in all animals at 35 days. These fragments were not attached to the TM surface and could be easily removed with a 1.6 mm aspirator. After the removal of the sponge collagen residue, the TM macrostructure within the perforation region was not visually distinct from that of the intact TM parts in five animals (83.3%), scarring was not visually determined (Figure 3A). In one case (16.7%), TM thickening and more pronounced scarring were noted at the site of the closed perforation. In spite of the observed tendency of a lower incidence of macroscopically identifiable scar changes in the TM in Group 2 (collagen use), the comparison of the groups using Fisher’s exact test did not find a statistical difference (two-sided *p*-value = 0.2424) (Figure 3B).

Endo-otoscopy of the 2d block of the experimental study (chronic perforations). In control Group 3, 14 days after tube removal and de-epidermization of the edges, no closure of the TM perforation was observed in any of the cases, and reactive phenomena were present in all the objects of study (*n* = 6) (Figure 3A). In experimental Group 4, 14 days after tube removal, de-epidermization of the edges and porous collagen graft implantation, the graft was found to be tightly adjoining the perforation region in all the animals. Its macrostructural integrity was preserved, and no repeated implantation was needed (Figure 3A).

After 35 days, in control Group 3, no cases of spontaneous closure of the TM perforation were observed. Endo-otoscopy visualized the TM with a central perforation of a toroidal shape with thickened edges; the epidermal layer was folded toward the tympanic cavity. Taking into account the obtained data, all the perforations were considered persistent (chronic) (Figure 3A). In experimental Group 4 (with the use of collagen), after 35 days, closure of the chronic TM perforations occurred in all the animals (six ears). Similar to experimental Group 2 (acute perforations closed with porous collagen), a small amount of unresorbed collagen material was present on the TM surface and in the deep parts of the external auditory channel in all the animals. It was not attached to the surrounding tissues and was removed with a vacuum aspirator. In all the animals, the TM was an integral structure, inflammatory changes were absent, a tender scar formed at the perforation site, which macroscopically was indistinguishable from the intact TM parts (Figure 3A).

### 3.3. Morphology Study

Intact TM. Throughout its length, the TM was very thin and slightly thickened only at the periphery, where it was attached to bone. The TM consisted of three layers: the outer layer, represented by multilayered basal epithelium, the middle layer of loose fibrous connective tissue, and the inner monolayer of cubic epithelium. The layered TM structure is especially clearly visible in phase contrast microscopy images (Figure 4A).

Control Group 1 (acute TM perforation). The TM was severely thickened over all its length and consisted of fibrous tissue formed by tightly packed longitudinally oriented collagen fibers with numerous fibroblasts between them. Moderate vascularization and notable lympho-macrophage infiltration with the addition of neutrophils was noted in the fibrous tissues, indicating inflammation (Figure 4A).

Experimental Group 2 (acute TM perforation). In this group, the TM was thin, with the normal structure, although it was thickened at some places (circa one third of its length). In most samples, the area of thickening consisted of two regions: a long solid region and a short loose one. The solid region presented fibrous tissue of a varying degree of maturation, which consisted of longitudinally oriented and tightly packed newly formed collagen fibers with a small number of fibroblasts in between and blood vessels. In more mature parts, the fibrous tissue forming the TM thinned, became denser and was almost devoid of blood vessels. In the loose regions of the thickening, defibration of the TM components was observed in the form of intertwined fiber bundles and thin newly formed loose collagen fibers. In rare cases, foci of newly formed cartilage were found. At times, residual fragments of the implanted collagen with a mesh-like structure were seen, permeated with cells. No inflammatory infiltration was observed in any of the samples of this group (Figure 4A). However, a semi-quantitative morphometric analysis of TM thickness, collagen fiber density, inflammation and vascularization (Table A1, Table A2, Table A3, Table A4 and Table A5) revealed no statistically significant differences between Groups 1 and 2.

Control Group 3 (chronic perforation). In all the samples, the region of perforation in the TM was open and not filled with fibrous tissue. The perforation edges were severely thickened and consisted of fibrous tissue with lympho-macrophage infiltration with the addition of neutrophils and defibrated TM components in the form of intertwined fiber bundles. In some parts, residual implanted collagen was found (Figure 4A).

Experimental Group 4 (chronic perforation). In most samples, about half of the TM length underwent fibrosis and was thickened by 4–5 times; in some samples, the TM was thickened along almost its entire length. The thickened TM was mostly represented by dense regions of fibrous tissue consisting of tightly packed collagen fibers oriented parallel to each other with a small number of fibroblasts between them. No inflammatory infiltration was present in the fibrous tissue, and the number of blood vessels was minimal. In rare cases, loosened parts with residual implanted collagen and small regions of newly formed cartilage were seen in this group. There was a trend (*p* = 0.07) towards a decrease in TM thickness and an increase in collagen fiber density in Group 4 compared to Group 3. Inflammation was significantly lower in Group 4 compared to Group 3 (*p* = 0.05). No significant differences were observed in the number of blood vessels.

### 3.4. The Study of the TM’s Vibrational Properties and Amplitude-Frequency Characteristics

Using the experimental setup with a laser fiber optic probe, we studied the vibrational properties and AFC of the intact TM and TM with acute and chronic perforations closed spontaneously or due the treatment (Group 1, 2, and 4). In Group 3, there were no cases of spontaneous closure of the TM perforation; therefore, it was not possible to measure the vibrational properties of the perforated TM. The measurements taken of the intact TM showed that the range of sensitivity determined by the level of average amplitudes was from ~Flow = 100 Hz for low frequencies to Fhigh~15–20 kHz for high frequencies. The highest amplitudes of the vibrational responses reached 60 dB via the relative scale in the logarithmic representation; thus, the reduction by 10–20 dB for amplitudes of the maximal responses in the case of experimental samples was assigned as significant, according to the conventional clinical indices of tone threshold audiometry.

In Group 1, we revealed a drop in sensitivity at middle and high frequencies (upper boundaries: 1, 2, 4 kHz) and a decrease in maximum response amplitudes by 30 dB, 25 dB, and 25 dB, respectively. The decrease in sensitivity at high frequencies and in maximum response amplitudes is typical for TM scars causing an increase in tissue stiffness. The vibrational TM characteristics in Group 2 at the upper and lower limits of the frequency range were similar to those measured for the intact TM in two samples (1. ~100–15,000 Hz, 65 dB; 2. ~100–20,000 Hz, 65 dB); one sample showed a slight decrease at the upper frequency limit (~100–12,000 Hz, 70 dB).

Analysis of the TM with the chronic perforation treated with the collagen matrix (Group 4) showed that the upper and lower boundaries of the frequency audibility range appeared to be not lower than 10 kHz and not higher than 100 Hz, respectively. The highest amplitudes for all samples were 80 dB, 65 dB, and 70 dB, i.e., the deviation from the amplitude of the intact TM did not exceed 15 dB.

## 4. Discussion

So far, a large number of scaffolds have been tested in the closure of both acute and chronic TM perforations [5]. A scaffold is a 3D network for the migration of cells and nutrients towards the TM defect in order to heal it. Each scaffold material has its specifics. To date, scaffolds are made of polymers and various decellularized tissues [38]. Decellularized tissues produced by the removal of cells from allo- and xenotransplants retain the mechanical and biological properties of the original transplant [20]. Allotransplants (AlloDerm, decellularized human dermis) and xenotransplants (UBM, decellularized bladder wall; SIS/SurgiSIS, small intestinal submucosa) [22,23,24,27,28,29,30,31,39] belong to such materials. At the same time, difficulties exist in the use of decellularized tissues (both allo- and xenotransplants) in clinical practice, related mainly to their safety and ethical issues [20,39]. Therefore, polymeric scaffolds have advantages in clinical applications.

Among the polymer materials, silk fibroin and chitosan were applied both in experimental studies and clinical practice, while alginate was tested only in the experimental setting [35,40,41,42,43,44,45,46,47]. Gelatin sponges impregnated with various growth factors or antibiotics are also actively used as scaffolds [5,48,49,50]. Recent clinical and experimental studies have shown that a gelatin sponge with an ofloxacin solution promotes TM healing [51]. The studies on bacterial cellulose, a PGS (polyglycerol-sebacate) synthetic biopolymer, a biologically modified collagen-immobilized PDMS scaffold are currently in progress [13,15,52,53,54,55].

In a number of studies, the use of collagen is described [13,15,18,56,57,58,59]. Collagen is an available, highly biocompatible natural polymer material, widely used in tissue engineering [60]. The scientific literature mentions the absence of the rejection reaction for collagen materials which makes them one of the best scaffolds for the closure of TM defects [60]. According to the research data, a collagen sponge has the advantage of accelerating the closure of a TM defect, as well as shrinking and spontaneous detachment after the defect is closed [57].

In our study, Achilles tendons from cattle were selected as a source of type I collagen for the following reasons: (1) this source is commercially available; (2) the native fibrillar structure of collagen is preserved after extraction; (3) in the course of SBA-EPD, the collagen suspension is stable, thus, the deposition is controllable [32]. The collagen membrane as a scaffold provides the stiffness and elasticity of tissues and, besides its contribution to the physico-mechanical characteristics of the collagen-based implant, it also affects regeneration and has a mechanical effect on cells. We have also developed the technology of collagen post-processing, which has allowed us to achieve porous and perforated collagen forms that better integrate into surrounding tissues. Porosity is known to increase the rate of biodegradation and to facilitate cell migration and infiltration; that is another advantage of the collagen matrix [5]. The purpose of this study was to assess the porous form of collagen in the repair of acute and chronic defects of the TM, along with its biodegradation. Our study on the application of a porous collagen graft in the closure of acute and chronic TM perforations in small laboratory animals demonstrates this material’s efficiency with all the perforations closed at the control times of observations. It is important that no notable scarring was observed at the sites of healed perforation in almost all the cases, in both acute and chronic perforation groups. The porous collagen membrane showed good biodegradability, with preserved macrostructural integrity on day 14 and almost complete resorption on day 35. The assessment of the influence of the membrane’s pores was out of the scope of this study in the restoration of the TM; the matrix has to be porous to ensure tissue ingrowth, so its porosity was considered an obligatory requirement to be orthotopically implanted [5,61,62]. In the future, it would be possible to combine the studied collagen graft with growth factors and cells. The application of 3D bioprinting appears especially promising, being one of the priorities of modern experimental studies.

The histological study of the tissue reaction to the collagen graft did not reveal signs of induced toxic effects or significant foreign body reaction. In particular, the tissues surrounding the implant were not infiltrated by leucocytes or macrophages in any of the samples. It should also be noted that the restored TM was represented mainly by dense regions of fibrous tissue consisting of tightly packed collagen fibers with a parallel orientation and a small number of fibroblasts between them. Thus, the structure of the restored TM fragment was close to that of an intact TM.

An important finding of the morphological study was the presence of small foci of newly formed cartilage in the groups after the application of porous collagen grafts. According to the literature data, latent progenitor cells were found in the TM, which were similar to endogenous stem cells of the skin. Knutsson et al. [63] found regions of regeneration where epithelial progenitor cells could be present. For this purpose, TM were harvested from five patients who underwent the removal of acoustic neuroma via the translabyrinthine approach. Using immunofluorescence and immunohistochemistry studies, integrin α6 was revealed in the basal layer of cornified epithelium in the regions of the umbo of the TM and the fibrous ring. Such markers as integrin and cytokeratin 19 were found in the same regions. Based on the obtained data, the authors assumed that potential progenitor stem cells in the TM are located in the regions of the umbo and the fibrous ring and along the malleus [63]. In the following studies, Korean researchers applied immunofluorescence and immunohistochemistry methods to detect markers of epithelial progenitor cells of the TM. Using experimental models of acute and chronic TM perforation, these authors demonstrated that latent stem cells of the TM may play a crucial role in its regeneration, both for acute and chronic defects. Thus, it is claimed by [64,65] that it is the stimulation of differentiation and proliferation of epithelial progenitor cells in the TM that plays the main role in the closure of both acute and chronic perforations. It is known that the foci of the metaplasia of fibrous connective tissues into cartilage tissue are frequently found in different tissues, including the larynx [66]. Accordingly, this finding is not negative when extrapolating the experimental conditions to clinical practice, since there is a long-standing practice of the application of autologous perichondrium and cartilage implants in tympanoplasty for defect closure and strengthening of the neo-TM [67,68].

The eardrum plays an important role as the first and highly sensitive structure that perceives sound energy, transforming it into corresponding mechanical vibrations of the further elements of the sound transmission chain [6]. The presence of many resonances, as well as damping of the membrane due to nearby resonances, determines the broadband transmission of sound energy to the cochlea [69]. Pathological changes, such as persistent chronic perforation, have a significant impact on the mechanical behavior of the TM [70]. In this study, the highly sensitive vibrometer with a laser fiber-optic probe revealed a decrease in the high frequency threshold in the restored TM with acute perforations up to 12 kHz and chronic perforations up to 10 kHz compared to the intact TM. Such a drop in sensitivity at high frequencies is characteristic of a small scar on the TM or an insignificant increase in the TM’s stiffness. This work has also shown the prospects of using an experimental setup to evaluate vibrations of the TM, which we plan to expand in future studies.

## 5. Conclusions

The application of bioengineered constructs for TM regeneration is aimed at the increased probability of successful perforation healing and at the minimization of the surgical intervention due to the stimulation of cell migration and proliferation. In the last few years, there has been a growing number of experimental studies on developing and testing new scaffolds, growth factors, cells and their combinations, and new clinical trials are being carried out. The use of a porous collagen graft for the closure of acute and chronic TM perforations in the experimental conditions appeared efficient, and opens an opportunity to implement this technique in clinical practice in the future. In contrast to conventional surgical techniques applied in the treatment of acute and chronic TM perforations, this method is faster, less invasive and reduces the time of rehabilitation. The obtained results could be used to design a pilot clinical trial focusing on closing acute and small persistent perforations. Moreover, the developed collagen matrix could serve as a scaffold or biopaper to fabricate TM bioequivalents. This approach seems to be promising for inducing the regeneration of large TM defects, including subtotal perforations or failure of the primary tympanoplasty.

## Figures and Tables

**Figure 1 polymers-16-00248-f001:**
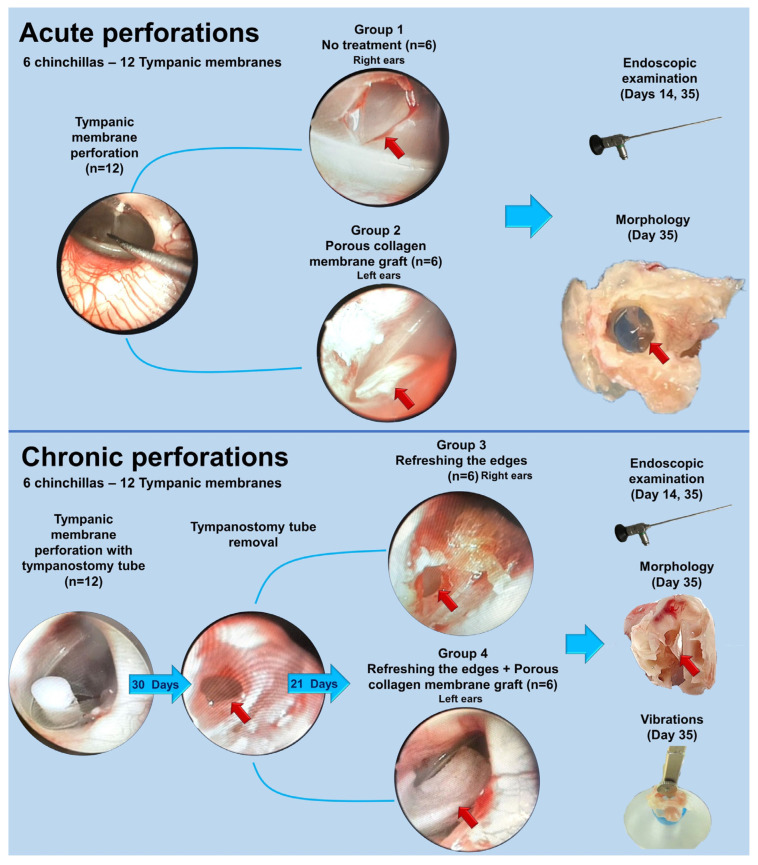
Experimental design. A porous collagen membrane was used, as a tested agent. The membrane was implanted into acute and chronic tympanic membrane model perforations in chinchillas on day 0 (acute perforations) and day 21 after the tympanostomy tube removal (chronic perforations). The endoscopic assessment was provided on day 14 and 35; the morphological study was performed on day 35, and, additionally for TM with chronic perforations, the vibrational study was performed on day 35. The stages of the acute and chronic tympanic membrane perforation model and implantation included: perforation with a microneedle and simultaneous collagen graft application (acute perforations) or insertion of the tympanostomy tube for chronic ones; tympanostomy tube removal on day 30, assessment of the perforation stability at 21 days and refreshing the perforation edges with or without collagen application. Endoscopy (0^0^). Red arrows indicate defects with or without the collagen graft and TM in the macropreparations.

**Figure 2 polymers-16-00248-f002:**
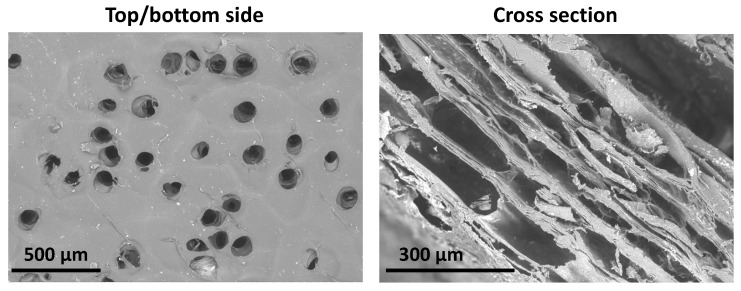
The microstructure of collagen matrix. The top and the bottom view and its cross section.

**Figure 3 polymers-16-00248-f003:**
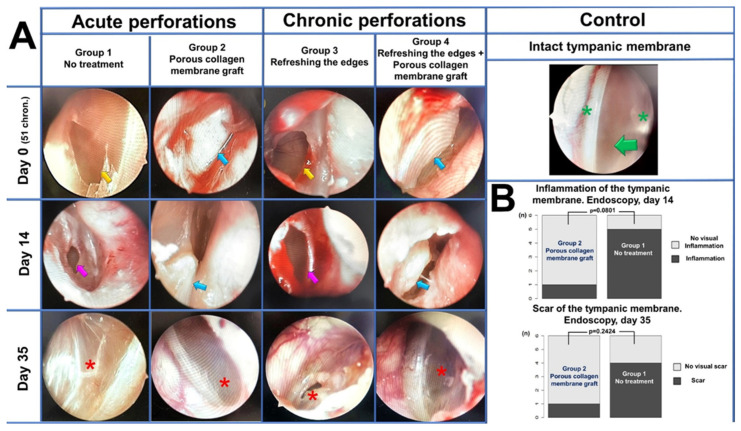
Endoscopic examination. (**A**) Endoscopic photos of the TM at the control points. Intact TM (green arrow) with fibrous ring and light reflex (green asterisks). Yellow arrows mark the edges of acute and chronic TM perforations on day 0. Purple arrows mark inflamed edges of acute and chronic TM perforations on day 14 in Groups 1 and 3. Blue arrows mark the collagen grafts on days 0, 14. Red asterisks mark the perforation site on day 35: Group 1—a prominent scar on the TM surface; Group 2—visually, the scar area is indistinguishable from an intact membrane; Group 3—residual chronic perforation; Group 4—the site of the regenerated chronic perforation. (**B**) TM with the endoscopic signs of inflammation on day 14 and prominent scars on day 35 in Groups 1 and 2; vertical scale shows the number of objects.

**Figure 4 polymers-16-00248-f004:**
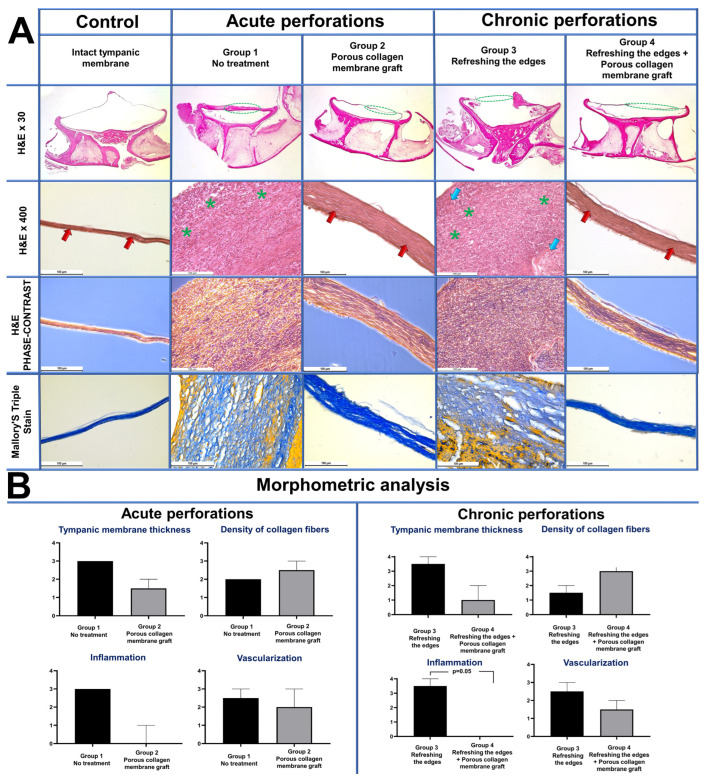
(**A**) Morphology on day 35. Light microscopy, H&E stain, magnification ×30. The region of injury is marked by a green dashed line: Control—an overall view of a thin normal TM; Group 1—severe TM thickening in the region of perforation and along the entire length; Group 2—thin TM, the region of the closed perforation is seen along the entire length and hardly distinguishable from the intact parts; Group 3—the TM defect region is not closed, severe TM thickening along the edge of perforation; Group 4—thin TM, the region of the closed perforation is seen along the entire length and hardly distinguishable from the intact parts. Light microscopy, H&E stain, magnification ×400. Control—layered structure of a normal TM, consisting of loose fibrous connective tissue located between the epithelial layers, collagen fibers of the fibrous layer are marked with red arrows; Group 1—formation of severe TM thickening in the perforation region due to fibrosis and pronounced inflammatory infiltration (green asterisks); Group 2—consolidation, thinning and maturation of fibrous tissue in the perforation region, scar formation, reduction in the number of inflammation cells and blood vessels, collagen fibers marked with red arrows; Group 3—the perforation edge is presented by fibrous tissue with the residual TM components (blue arrows) and pronounced inflammatory infiltration (green asterisks); Group 4—thin dense scar consisting of mature fibrous tissue with minimal vascularization and absence of inflammatory infiltration. Phase-contrast microscopy, H&E stain, magnification ×400; Control—layered structure of an intact TM is clearly seen; Group 1—in the region of thickening, the fibrous tissue consists of longitudinally oriented collagen fibers and numerous cells between them; Group 2—thin scar is presented by densely and longitudinally packed collagen fibers and rare cells and blood vessels between them; Group 3—fibrous tissue with intertwined collagen fibers, numerous cells and residual TM components; Group 4—thin scar of densely and longitudinally packed collagen fibers with a minimal number of cells and blood vessels. Light microscopy, Mallory’s triple stain, magnification ×400. Collagen fibers are colored blue, elastic fibers, blood vessels and cells of the inflammatory infiltrate are colored yellow. Control—normal thin TM consisting of numerous longitudinally packed collagen fibers; Group 1—numerous cells of inflammatory infiltration and blood vessels are located among collagen fibers of fibrous tissue in the region of TM thickening; Group 2—scar of fibrous tissue with minimal vascularization and inflammatory infiltration in the TM perforation region; Group 3—pronounced inflammatory infiltration and vascularization in the fibrous tissue near the edge of unclosed TM perforation; Group 4—thin dense scar in the perforation region consisting of mature fibrous tissue with minimal vascularization and absence of inflammatory infiltration. (**B**) Morphometric analysis. Vertical scale displays morphological scores, Median ± CI.

**Table 1 polymers-16-00248-t001:** Physical parameters of collagen matrices.

Matrix	Dry Thickness, µm	Swelling, %	Shrinkage Temperature, °C
Initial SBA-EPD collagen membrane	250 ± 30	410 ± 50	55 ± 1
Porous collagen membrane	440 ± 20	860 ± 70	55 ± 1

Abbreviations: SBA-EPD—semipermeable barrier-assisted electrophoretic deposition.

**Table 2 polymers-16-00248-t002:** Mechanical properties of collagen matrices.

Matrix	Young’s Modulus, MPa	Strain at Failure, %
Initial SBA-EPD collagen membrane	15 ± 2	67 ± 4
Porous collagen membrane	3 ± 1	53 ± 8

Abbreviations: SBA-EPD—semipermeable barrier-assisted electrophoretic deposition matrices.

## Data Availability

The relevant data generated and analyzed in the current study are available from the corresponding author upon reasonable request.

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
