# Peer review of "Collagen Matrix to Restore the Tympanic Membrane: Developing a Novel Platform to Treat Perforations"

_polymers, 2024, doi:10.3390/polym16020248_

Round 1
Reviewer 1 Report
Comments and Suggestions for Authors
General comments:
I) Porous collagen graft was tested to close acute and chronic tympanic membrane perforations in a chinchilla model. Otoscopic assessments were conducted on days 14 and 35, with histological evaluations and tympanic membrane vibrational determinations. The findings appear to be incomplete due to a lack of comparisons with other graft compounds as well as with SBA-EPD collagen membrane.
II) Several statements lacked references.
III) The number of chinchilla 12 or 24 (4 groups of 6 ?) need to be checked.
IV) The methodological part, especially buffer composition and pH are insufficiently delineated (see minor comments 9, 10, 11, 12, 14 and 21 for further details).
V) It was unclear why tympanic membrane vibrational determinations were performed only in group 4 and were not compared with other groups.
VI) Several parts of the discussion could be merged into the discussion.
VII) Further experiments are needed to support the hypothesis that “the foci of chondrogenesis may be, to some extent, related to the porous collagen graft’s function as a scaffold for progenitor cells of the TM forming a favorable matrix for their migration and proliferation not only to fibroblasts but also to chondroblasts”
Minor comments:
1) Introduction, p1: Add references to support “Perforation is considered persistent if it has not closed on its own in three months.”
2) Introduction, p1: Add references to support “The closure of the TM perforation is aimed at the restoration of not only its integrity but also its mechanic and acoustical properties playing a crucial role in the sound transmission.”
3) Introduction, p2: Add references to support “The efficiency of the surgical treatment de-pends on many factors, including the applied transplant’s thickness, the surgeon’s experience, and the preparation of the defect edges prior to the procedure.”
4) Introduction, p2: Replace “new” by alternate or another word in “Development of new alternative ways to restore the TM defects, which would al-low avoiding problems inherent in the surgical treatment, is an important problem in the modern medicine.”
5) Introduction, p2: Add references to support “A Significant attention has been paid lately to the methods of regenerative medicine in the closure of TM perforations”
6) Introduction, p2: Add references to support “To date, many scaffolds have been tested in the closure of both acute and chronic TM perforations.”
7) Introduction, p2: Add references to support “Collagen-based materials are used frequently for this purpose, with some differences in their physicochemical properties”
8) Introduction, p2: Add references to support “The developed technique has been shown to produce defect-free uniform membranes with no toxicity revealed in the in vitro and in vivo studies, and with a good biodegradation profile.”
9) Materials and Methods, Creation of the collagen matrix, Preparation of a collagen suspension, p2: Indicate pH in “The tendon fragments underwent four 12-h- long treatments with 0.5 M NaCl, followed by homogenization in 0.8 M acetic acid.”
10) Materials and Methods, Creation of the collagen matrix, Preparation of a collagen suspension, p2: Replace % by units (ie mg:mL) in “Pepsin (0.1%) was added to the resulting suspension, the hydrolysis proceeded for 2 days.”
11) Materials and Methods, Creation of the collagen matrix, Preparation of a collagen suspension, p2: Replace % by concentration “Then, the pH was increased to 7.5 using 1 M NaOH to stop the hydrolysis, and the collagen suspension was precipitated with a 12% NaCl solution.”
12) Materials and Methods, Creation of the collagen matrix, Preparation of a collagen suspension, p2-3: Indicate pH in both solutions. It was unclear why dialysis were performed at different pH in “The deposited collagen was redisolved in 0.8 M acetic acid and dialyzed against 0.5 M acetic acid for 3 days, the dialysis solution being replaced every day.”
13) Materials and Methods, Creation of the collagen matrix, Preparation of a collagen membrane by SBA-EPD, Title, p3: Replace SBA-EPD by its full name in the title.
14) Materials and Methods, Collagen matrix characterization, p3: Phosphate tends to precipitate most polyvalent cations and is either a metabolite or an inhibitor in many systems (Good, et al., Hydrogen ion buffers for biological research. Biochemistry. 1966;5:467-477.). Justify the use of PBS buffer, indicate composition and pH to support “To measure the swelling values,the dried samples of collagen matrixwere first 1weighed, then they were placed in PBS at 4 °C for 8h.”
15) Materials and methods, TM regeneration study, Title, p3: Replace TM by its full name in the title.
16) Materials and methods, TM regeneration study, Study design, p3: Check the number of chinchillas. Total number of chinchillas were 24 according to figure 1 (four groups of six), while 12 chinchillas were used in “In this study, 12 male chinchillas were used, of 6 months of age and 500-700 g of weight, from the central animal facility of the Sechenov University. In this study, 12 male chinchillas were used, of 6 months of age and 500-700 g of 1weight, from the central animal facility of the Sechenov University.”
17) Materials and methods, TM regeneration study, Study design, p3: Explain why there are two groups acute and chronic to support “By its design, it was a controlled experimental study with parallel groups (Figure 1). The experiments were divided into two blocks based on the model of the TM perforation: 1) acute; 2) chronic.”
18) Materials and methods, TM regeneration study, Study design, p5: Check the number of chinchillas. Total number of chinchillas were 24 according (four groups of six), while 12 chinchillas were used in “In the first block (acute perforations), a bilateral TM defect was created surgically in 6 animals. Right-side perforations were left for observation only (control Group 1, n=6). Left-side perforations were closed using a porous collagen graft immediately after the creation (experimental Group 2, n=6).” And in “Then, the chronic perforation edge was deepidermized. On the right side, the closure of perforation was not performed (control Group 3, n=6). On the left side, the perforations were closed with a porous collagen graft (experimental Group 4, n=6).”
19) Materials and methods, TM regeneration study, Surgical procedures, p5: Check the number of chinchillas. Total number of chinchillas were 24 according (four groups of six), while 12 chinchillas were used in “In the first block (acute perforations), a bilateral TM defect was created surgically in 6 animals in the whole paragraph.
20) Materials and methods, Endo-Otoscopy, p6: Indicate number of independent measurements to support “To estimate the statistical significance of the groups by these signs, Fisher’s exact test was used, with the threshold value of p=0.1.”
21) Materials and methods, Morphology study, p6: Replace % by units, pH and composition of the buffer in “Tissues fixed in 10% neutral buffered formalin were then decalcified, dehydrated and embedded in paraffin blocks. 4 µm-thick sections were stained with hematoxylin-eosin and with Mallory's trichrome stain and studied with a Leica DM 4000 B LED universal microscope equipped with a Leica DFC 7000 T camera under the control of the LAS V4.8 software (Leica Microsystems, Germany).”
22) Materials and methods, Endo-Otoscopy, p6: Indicate number of independent measurements to support “Low-magnification-photographs were obtained with a Bresser USB microscope (Bresser, Germany).”
23) Materials and methods, Morphometric analysis, p6: “In each sample in the course of the collagen membrane’s biological integration in the TM perforations, the signs of inflammation and regeneration (the TM thickness, density of collagen fibers, inflammatory infiltration, vascularization) were estimated by a 5-point scale (0 – no, 4 – highest intensity) (Tables A1-A4).”
24) Materials and methods, Study of vibrational properties and the amplitude-frequency characteristic of the TM, Title, p6: replace TM by its full name in the title.
25) Materials and methods, Study of vibrational properties and the amplitude-frequency characteristic of the TM, p7: Indicate number of independent measurements to “Using this technique, the measurements were conducted on 3 intact TM and 3 experimental TM from Group 4 with chronic perforations closed 35 days after the implantation of the sponge collagen grafts.”
26) Statistical Analysis, p7 : Indicate which measurements were subjected to. A statistical analysis and the number of independent measurements to support “The differences were estimated using the Kruskal–Wallis test with the Dunn’s test for multiple comparisons. P ≤ 0.05 were considered as significant. The results of the statistical analysis were presented as histograms of median values and the 95% confidence interval (CI).”
27) Results, Collagen matrix characteristics, p7 : Indicate number of independent measurements in Table-1 legend to support “The thickness and swelling values of the porous collagen membranes differed from those of the initial SBA-EPD matrices (Table 1), however their shrinkage temperatures were identical. ”
28) Results, Collagen matrix characteristics, p8 : Indicate number of independent measurements in Table-2 legend to support “When testing the mechanical properties, as shown in Table 2, the Young’s modulus and strain at failure of porous collagen form decreased compared to initial SBA-EPD matrices by 12 MPa and 14%, respectively.”
29) Results, Endo-Otoscopy, p8 : It was unclear how was determined the inflammation to support “When comparing Groups 1 and 2 by the presence of inflammatory changes, the differences were significant (Fisher’s exact test, two-sided p-value = 0.0801), being less pronounced in the Group 2 with the application of a collagen graft (Fig- ure 3B). ”
30) Results, Morphology study, p12 : Morphological analysis of group 2 and group 4 appears to be similar. This shall be commented to support “Studying the thickness of the TM and the density of collagen fibres, no statistically significant difference was found between groups 1 and 2. However, there was a trend (p = 0.07) towards a decrease in TM thickness and an increase in collagen fibre density in group 4 compared to group 3. Inflammation was significantly lower in group 4 compared to group 3 (p=0.05), but there was no difference between groups 1 and 2. No significant differences were observed in the number of vessels in all groups.”
31) Results, The study of the TM’s vibrational properties and amplitude-frequency characteristics, p12: It was unclear if the vibrational properties in group 4 were similar than in those observed in other groups to support “The data of the measurements on the TM with chronic perforations closed after the treatment with porous collagen grafts showed that the upper and lower boundaries of the frequency audibility range appeared not lower than 10 kHz and not higher than 100 Hz, respectively. The highest amplitudes for three samples were 80 dB, 65 dB, and 70 dB, i.e.the deviation from the amplitude of intact TM did not exceed 15 dB.”
32) Discussion, p12: Merge into the introduction, and add references to support “Transforming a sound wave into a mechanical vibration, the TM plays a key role in the sound conduction, since it represents a peculiar border between the middle ear and inner ear.”
33) Discussion, p12: Merge into the introduction, and add references to support “The dynamic properties of the TM are anatomically determined by the mobility of the chain of the auditory ossicles, as well as by the state of the TM itself, its thickness 4and structure.”
34) Discussion, p12: Merge into the introduction, and add references to support “The correct TM functioning directly depends on its vibrational competence which is determined by the physical properties of its middle fibrous layer.”
35) Discussion, p12: Merge into the introduction, and add references to support “In turn, the middle layer is represented by two layers of collagen fibers, radially and circularly oriented.”
36) Discussion, p12: Merge into the introduction “These collagen fibers are responsible for the viscoelastic properties of the TM, which, in turn, characterize the TM mechanics [5,6].”
37) Discussion, p12: Merge into the introduction “The perforation of the TM leads to changes in its mechanical properties and hearing impairment. Therefore, in the field of otolaryngology, the search for materials with structures and mechanical properties similar to those of the TM, and their use as scaffolds to improve the regeneration of defects and restore the vibrational competence of the TM, remains a crucial issue.”
38) Discussion, p13: Merge into the introduction, and add references to support “Since 1990s, with the development of tissue engineering, synthetic materials have demonstrated a great potential in the TM restoration.”
39) Discussion, p12: Merge into the introduction “The use of tissue engineered scaffolds provides significant advantages such as the reduction of the time of surgery and minimizing the surgical tissue damage, reduction of economic expenses, increase in the percentage of a positive morphofunctional result [5,7].”
40) Discussion, p13: Merge into the introduction, and add references to support “In spite of the TM’s inherent regenerative capacity, a support under the growing epithelial layer is required for its restoration.”
41) Discussion, p13: Merge into the introduction, and add references to support “ Different scaffolds or transplants are applied in the TM bioengineering, which serve as supports to facilitate migration of cells and nutrients towards the perforation.”
42) Discussion, p13: Merge into the introduction“ Opposite to conventional transplants retained in vivo, a bioengineered construct is meant to degrade with time and to be replaced with a new tissue [8]. We view the perfect purpose of tissue engineering techniques in the restoration of the three-layered TM structure which is crucial for sustaining its mechano-acoustic properties.”
43) Discussion, p13: Merge into the introduction, and add references to support “ Various scaffolds, stem cells and growth factors are used for the TM regenertion.”
44) Discussion, p13: Merge into the introduction “The above materials have been applied both separately and in various combinations, according to the data of different studies [8–10].”
45) Discussion, p13: Merge into the “ All these components play a vital role both for cell differentiation, proliferation and migration and for the creation of a proper biochemical microenvironment to prepare and stimulate the tissue healing [5,11,12].”
46) Discussion, p13: Add references to support “By now a large number of scaffolds have been tested in the closure of both acute and chronic TM perforations”
47) Discussion, p14: It is unclear if Initial SBA-EPD collagen membrane could induce similar effects than the porous collagen application in “It is important that no notable scarring was observed at the sites of healed perforation in almost all the cases of porous collagen application, in both acute and chronic perforation groups. The material showed a good biodegradability with the preserved macrostructural integrity on day 14 and almost complete resorption on day 35.”
48) Discussion, p14: It is unclear how lack of inflammation was monitored to support “Of interest are the results of the histological study. First of all, no inflammatory infiltration was observed in all the samples after the collagen graft application. This finding indicates that the material was biologically compatible with the cellular and tissue structure of the TM and, moreover, prevented the addition of a bacterial infection of the middle ear. “
49) Discussion, p14: Rephrase. There were only measurements on group 4 and it was not compared with other groups to adequately support “The mentioned data allow one to assume that, besides the potential repair of the fibrous layer, the TM’s vibrational properties also return to their normal state, however, additional studies are needed.”
50) Discussion, p14: Further experiments are needed to confirm “The above data make it possible to assume that it is the stimulation of differentiation and proliferation of epithelial progenitor cells in the TM that plays the main role in the closure of both acute and chronic perforations [46,47].”
51) Discussion, p14: Further experiments are needed to confirm “ “Thus, the presence of the foci of chondrogenesis may be, to some extent, related to the porous collagen graft’s function as a scaffold for progenitor cells of the TM forming a favorable matrix for their migration and proliferation not only to fibroblasts but also to chondroblasts.”
52) Discussion, p14: Add references to support “Accordingly, this finding is not negative when extrapolating the experimental conditions to clinical practice, since there is a long-standing practice of application of autologous perichondrium and cartilage implants in tympanoplasty for the defect closure and strengthening of the neo-TM.”
53) Discussion, p14: Add references to support “The eardrum plays an important role as the first and highly sensitive structure that perceives sound energy, transforming it into corresponding mechanical vibrations of further elements of the sound transmission chain.”
54) Discussion, p14: Add references to support “Pathological changes, persistent chronic perforation has a significant impact on the mechanical behavior of the TM.”
55) Discussion, p14-15: Apparently group4 was not compared with other groups. Rephrase “In this study, the highly sensitive vibrometer with a laser fiber-optic probe revealed decrease in high frequency threshold in the restored tympanic membranes up to 10 kHz comparing to the intact membranes”
Author Response
Reviewer 1
General comments:
I. Porous collagen graft was tested to close acute and chronic tympanic membrane perforations in a chinchilla model. Otoscopic assessments were conducted on days 14 and 35, with histological evaluations and tympanic membrane vibrational determinations. The findings appear to be incomplete due to a lack of comparisons with other graft compounds as well as with SBA-EPD collagen membrane.
ANSWER: The matrix porosity is one of the main parameters ensuring better connective tissue ingrowth and vascularization and, therefore, quality of regeneration (10.1088/1758-5090/ac88a1, 10.1089/ten.tea.2010.0571). Thus, this parameter is considered to be obligatory that makes the developed matrix capable to be used to restore the TM perforations.
II. Several statements lacked references.
ANSWER: The references was added.
III. The number of chinchilla 12 or 24 (4 groups of 6 ?) need to be checked.
ANSWER: The description was carefully checked. We used twelve chinchillas and formed four groups (six tympanic membranes (TM) per each; n=6 shows six perforated TM per each group). Acute perforations were performed bilaterally (right and left) in six animals (twelve TM): right-sided TM (n=6) served as a control (group 1); left-sided TM (n=6) were closed with the collagen membrane (group 2). The groups with chronic TM perforations were formed similarly.
IV. The methodological part, especially buffer composition and pH are insufficiently delineated (see minor comments 9, 10, 11, 12, 14 and 21 for further details).
ANSWER: The requested information was added.
V. It was unclear why tympanic membrane vibrational determinations were performed only in group 4 and were not compared with other groups.
ANSWER: Sections 2.7 and 3.4 were extended. Data regarding group 1 and 2 were included. In group 3, there were no cases of the spontaneous closure of the TM perforation; therefore, it was not possible to measure the vibrational properties of the perforated TM.
Section 2.7: “Using this technique, we measured three intact TM and experimental ones: group 1 – acute perforation, control, n=3 (spontaneous closure in 35 days); group 2 – acute perforation, porous collagen membrane, n=3 (closure in 35 days); and group 4 – chronic perforation, porous collagen membrane, n=3 (closure in 35 days). In group 3, there were no cases of the spontaneous closure of the TM perforation; therefore, it was not possible to measure the vibrational properties of the perforated TM. The number of independent measurements was five”.
Section 3.4: “Using the experimental setup with a laser fiber optic probe, we studied the vibrational properties and AFC of the intact TM and TM with acute and chronic perforations closed spontaneously or due the treatment (group 1, 2, and 4). In group 3, there were no cases of the spontaneous closure of the TM perforation; therefore, it was not possible to measure the vibrational properties of the perforated TM…
In group 1, we revealed a drop in the sensitivity at middle and high frequencies (upper boundaries: 1, 2, 4 kHz) and a decrease in maximum response amplitudes by 30 dB, 25 dB, and 25 dB, respectively. The decrease in sensitivity at high frequencies and in maximum response amplitudes is typical for TM scars causing an increase in tissue stiffness. The vibrational TM characteristics in group 2 at the upper and lower limits of the frequency range were similar to those measured for the intact TM in two samples (1. ~ 100 Hz – 15000 Hz, 65 dB; 2. ~100 Hz-20000 Hz, 65 dB); one sample showed a slight decrease at the upper frequency limit (~100 Hz – 12000 Hz, 70 dB).
Analysis of the TM with the chronic perforation treated with the collagen matrix (group 4) showed that the upper and lower boundaries of the frequency audibility range appeared to be not lower than 10 kHz and not higher than 100 Hz, respectively. The highest amplitudes for all samples were 80 dB, 65 dB, and 70 dB, i.e. the deviation from the amplitude of the intact TM did not exceed 15 dB”.
VI. Several parts of the discussion could be merged into the discussion.
ANSWER: The text was corrected.
VII. Further experiments are needed to support the hypothesis that “the foci of chondrogenesis may be, to some extent, related to the porous collagen graft’s function as a scaffold for progenitor cells of the TM forming a favorable matrix for their migration and proliferation not only to fibroblasts but also to chondroblasts”
ANSWER: The statement was removed.
Minor comments:
1. Introduction, p1: Add references to support “Perforation is considered persistent if it has not closed on its own in three months.”
ANSWER: The reference was added.
2. Introduction, p1: Add references to support “The closure of the TM perforation is aimed at the restoration of not only its integrity but also its mechanic and acoustical properties playing a crucial role in the sound transmission.”
ANSWER: The reference was added.
3. Introduction, p2: Add references to support “The efficiency of the surgical treatment de-pends on many factors, including the applied transplant’s thickness, the surgeon’s experience, and the preparation of the defect edges prior to the procedure.”
ANSWER: The reference was added.
4. Introduction, p2: Replace “new” by alternate or another word in “Development of new alternative ways to restore the TM defects, which would al-low avoiding problems inherent in the surgical treatment, is an important problem in the modern medicine.”
ANSWER: The word was removed.
5. Introduction, p2: Add references to support “A Significant attention has been paid lately to the methods of regenerative medicine in the closure of TM perforations”
ANSWER: The reference was added.
6. Introduction, p2: Add references to support “To date, many scaffolds have been tested in the closure of both acute and chronic TM perforations.”
ANSWER: The reference was added.
7. Introduction, p2: Add references to support “Collagen-based materials are used frequently for this purpose, with some differences in their physicochemical properties”
ANSWER: The reference was added.
8. Introduction, p2: Add references to support “The developed technique has been shown to produce defect-free uniform membranes with no toxicity revealed in the in vitro and in vivo studies, and with a good biodegradation profile.”
ANSWER: The reference was added.
9. Materials and Methods, Creation of the collagen matrix, Preparation of a collagen suspension, p2: Indicate pH in “The tendon fragments underwent four 12-h- long treatments with 0.5 M NaCl, followed by homogenization in 0.8 M acetic acid.”
ANSWER: The requested information was included.
10. Materials and Methods, Creation of the collagen matrix, Preparation of a collagen suspension, p2: Replace % by units (ie mg:mL) in “Pepsin (0.1%) was added to the resulting suspension, the hydrolysis proceeded for 2 days.”
ANSWER: The units were changed.
11. Materials and Methods, Creation of the collagen matrix, Preparation of a collagen suspension, p2: Replace % by concentration “Then, the pH was increased to 7.5 using 1 M NaOH to stop the hydrolysis, and the collagen suspension was precipitated with a 12% NaCl solution.”
ANSWER: The units were changed.
12. Materials and Methods, Creation of the collagen matrix, Preparation of a collagen suspension, p2-3: Indicate pH in both solutions. It was unclear why dialysis were performed at different pH in “The deposited collagen was redisolved in 0.8 M acetic acid and dialyzed against 0.5 M acetic acid for 3 days, the dialysis solution being replaced every day.”
ANSWER: The requested information was included. Initially lower pH (2.7) was required for better collagen dissolution, while the dialysis allowed for adjusting pH to optimal values for further SBA-EPD (pH 2.9).
13. Materials and Methods, Creation of the collagen matrix, Preparation of a collagen membrane by SBA-EPD, Title, p3: Replace SBA-EPD by its full name in the title.
ANSWER: The abbreviation was removed.
14. Materials and Methods, Collagen matrix characterization, p3: Phosphate tends to precipitate most polyvalent cations and is either a metabolite or an inhibitor in many systems (Good, et al., Hydrogen ion buffers for biological research. Biochemistry. 1966;5:467-477.). Justify the use of PBS buffer, indicate composition and pH to support “To measure the swelling values,the dried samples of collagen matrixwere first 1weighed, then they were placed in PBS at 4 °C for 8h.”
ANSWER: The required information was added. The PBS was purchased from Sigma Aldrich (Germany) and was ready to be used. The collagen in the membrane is in the “precipitated” state, so no changes will occur after its placing into PBS except hydration, which extent is measured in the “swelling” test. Testing collagen samples swelling in PBS is a standard procedure described elsewhere (10.1016/j.biomaterials.2003.09.066).
15. Materials and methods, TM regeneration study, Title, p3: Replace TM by its full name in the title.
ANSWER: The abbreviation was removed.
16. Materials and methods, TM regeneration study, Study design, p3: Check the number of chinchillas. Total number of chinchillas were 24 according to figure 1 (four groups of six), while 12 chinchillas were used in “In this study, 12 male chinchillas were used, of 6 months of age and 500-700 g of weight, from the central animal facility of the Sechenov University. In this study, 12 male chinchillas were used, of 6 months of age and 500-700 g of 1weight, from the central animal facility of the Sechenov University.”
ANSWER: The description was carefully checked, and Figure 1 was corrected. We used twelve chinchillas and formed four groups (six tympanic membranes (TM) per each; n=6 shows six perforated TM per each group). Acute perforations were performed bilaterally (right and left) in six animals (twelve TM): right-sided TM (n=6) served as a control (group 1); left-sided TM (n=6) were closed with the collagen membrane (group 2). The groups with chronic TM perforations were formed similarly.
17. Materials and methods, TM regeneration study, Study design, p3: Explain why there are two groups acute and chronic to support “By its design, it was a controlled experimental study with parallel groups (Figure 1). The experiments were divided into two blocks based on the model of the TM perforation: 1) acute; 2) chronic.”
ANSWER: Figure 1 and the sentence were corrected. Acute perforations were performed bilaterally (right and left) in six animals (twelve TM): right-sided TM (n=6) served as a control (group 1); left-sided TM (n=6) were closed with the collagen membrane (group 2). The groups with chronic TM perforations were formed similarly. Thus, acute and chronic perforations form two different blocks of groups in order to show the applicability of the developed membrane in two pathological types.
18. Materials and methods, TM regeneration study, Study design, p5: Check the number of chinchillas. Total number of chinchillas were 24 according (four groups of six), while 12 chinchillas were used in “In the first block (acute perforations), a bilateral TM defect was created surgically in 6 animals. Right-side perforations were left for observation only (control Group 1, n=6). Left-side perforations were closed using a porous collagen graft immediately after the creation (experimental Group 2, n=6).” And in “Then, the chronic perforation edge was deepidermized. On the right side, the closure of perforation was not performed (control Group 3, n=6). On the left side, the perforations were closed with a porous collagen graft (experimental Group 4, n=6).”
ANSWER: The description was carefully checked. We used twelve chinchillas and formed four groups (six tympanic membranes (TM) per each; n=6 shows six perforated TM per each group). Acute perforations were performed bilaterally (right and left) in six animals (twelve TM): right-sided TM (n=6) served as a control (group 1); left-sided TM (n=6) were closed with the collagen membrane (group 2). The groups with chronic TM perforations were formed similarly.
19. Materials and methods, TM regeneration study, Surgical procedures, p5: Check the number of chinchillas. Total number of chinchillas were 24 according (four groups of six), while 12 chinchillas were used in “In the first block (acute perforations), a bilateral TM defect was created surgically in 6 animals in the whole paragraph.
ANSWER: The description was carefully checked. We used twelve chinchillas and formed four groups (six tympanic membranes (TM) per each; n=6 shows six perforated TM per each group). Acute perforations were performed bilaterally (right and left) in six animals (twelve TM): right-sided TM (n=6) served as a control (group 1); left-sided TM (n=6) were closed with the collagen membrane (group 2). The groups with chronic TM perforations were formed similarly.
20. Materials and methods, Endo-Otoscopy, p6: Indicate number of independent measurements to support “To estimate the statistical significance of the groups by these signs, Fisher’s exact test was used, with the threshold value of p=0.1.” Укаждого объекта по 3 раза.
ANSWER: The requested information was added.
21. Materials and methods, Morphology study, p6: Replace % by units, pH and composition of the buffer in “Tissues fixed in 10% neutral buffered formalin were then decalcified, dehydrated and embedded in paraffin blocks. 4 µm-thick sections were stained with hematoxylin-eosin and with Mallory's trichrome stain and studied with a Leica DM 4000 B LED universal microscope equipped with a Leica DFC 7000 T camera under the control of the LAS V4.8 software (Leica Microsystems, Germany).”
ANSWER: The requested information was included.
22. Materials and methods, Endo-Otoscopy, p6: Indicate number of independent measurements to support “Low-magnification-photographs were obtained with a Bresser USB microscope (Bresser, Germany).”
ANSWER: A Bresser USB microscope (Bresser, Germany) was used to obtain whole slide images of histological slides which can be found at the high row of Figure 4 titled “H&E x30”. We corrected the sentence. “Low-magnification images of histological slides were obtained with a Bresser USB microscope (Bresser, Germany).”
23. Materials and methods, Morphometric analysis, p6: “In each sample in the course of the collagen membrane’s biological integration in the TM perforations, the signs of inflammation and regeneration (the TM thickness, density of collagen fibers, inflammatory infiltration, vascularization) were estimated by a 5-point scale (0 – no, 4 – highest intensity) (Tables A1-A4).”
ANSWER: The detailed information is presented in Tables B1-B4 (Appendix B).
24. Materials and methods, Study of vibrational properties and the amplitude-frequency characteristic of the TM, Title, p6: replace TM by its full name in the title.
ANSWER: The abbreviation was removed.
25. Materials and methods, Study of vibrational properties and the amplitude-frequency characteristic of the TM, p7: Indicate number of independent measurements to “Using this technique, the measurements were conducted on 3 intact TM and 3 experimental TM from Group 4 with chronic perforations closed 35 days after the implantation of the sponge collagen grafts.”
ANSWER: The requested information was added.
26. Statistical Analysis, p7 : Indicate which measurements were subjected to. A statistical analysis and the number of independent measurements to support “The differences were estimated using the Kruskal–Wallis test with the Dunn’s test for multiple comparisons. P ≤ 0.05 were considered as significant. The results of the statistical analysis were presented as histograms of median values and the 95% confidence interval (CI).”
ANSWER: The sentence was corrected.
27. Results, Collagen matrix characteristics, p7 : Indicate number of independent measurements in Table-1 legend to support “The thickness and swelling values of the porous collagen membranes differed from those of the initial SBA-EPD matrices (Table 1), however their shrinkage temperatures were identical. ”
ANSWER: The requested information was added.
28. Results, Collagen matrix characteristics, p8 : Indicate number of independent measurements in Table-2 legend to support “When testing the mechanical properties, as shown in Table 2, the Young’s modulus and strain at failure of porous collagen form decreased compared to initial SBA-EPD matrices by 12 MPa and 14%, respectively.”
ANSWER: The requested information was included.
29. Results, Endo-Otoscopy, p8 : It was unclear how was determined the inflammation to support “When comparing Groups 1 and 2 by the presence of inflammatory changes, the differences were significant (Fisher’s exact test, two-sided p-value = 0.0801), being less pronounced in the Group 2 with the application of a collagen graft (Fig- ure 3B). ”
ANSWER: While comparing groups 1 and 2, we analyzed the presence or absence of the TM and middle ear inflammation signs: hyperemia, edema, mucosa’s blood vessels, mucous purulent exudate, granulation, and size of the perforation. The presence or absence of these signs were assigned qualitative values of “1” or “0”, respectively (Section 2.4).
30. Results, Morphology study, p12 : Morphological analysis of group 2 and group 4 appears to be similar. This shall be commented to support “Studying the thickness of the TM and the density of collagen fibres, no statistically significant difference was found between groups 1 and 2. However, there was a trend (p = 0.07) towards a decrease in TM thickness and an increase in collagen fibre density in group 4 compared to group 3. Inflammation was significantly lower in group 4 compared to group 3 (p=0.05), but there was no difference between groups 1 and 2. No significant differences were observed in the number of vessels in all groups.”
ANSWER: We divided the paragraph and moved the statistical statements after the relevant morphological descriptions.
31. Results, The study of the TM’s vibrational properties and amplitude-frequency characteristics, p12: It was unclear if the vibrational properties in group 4 were similar than in those observed in other groups to support “The data of the measurements on the TM with chronic perforations closed after the treatment with porous collagen grafts showed that the upper and lower boundaries of the frequency audibility range appeared not lower than 10 kHz and not higher than 100 Hz, respectively. The highest amplitudes for three samples were 80 dB, 65 dB, and 70 dB, i.e.the deviation from the amplitude of intact TM did not exceed 15 dB.”
ANSWER: The description was expanded. In group 3, there were no cases of the spontaneous closure of the TM perforation; therefore, it was not possible to measure the vibrational properties of the perforated TM.
32. Discussion, p12: Merge into the introduction, and add references to support “Transforming a sound wave into a mechanical vibration, the TM plays a key role in the sound conduction, since it represents a peculiar border between the middle ear and inner ear.”
ANSWER: The sentence was transferred; the references were added.
33. Discussion, p12: Merge into the introduction, and add references to support “The dynamic properties of the TM are anatomically determined by the mobility of the chain of the auditory ossicles, as well as by the state of the TM itself, its thickness and structure.”
ANSWER: The sentence was transferred; the references were added.
34. Discussion, p12: Merge into the introduction, and add references to support “The correct TM functioning directly depends on its vibrational competence which is determined by the physical properties of its middle fibrous layer.”
ANSWER: The sentence was transferred; the references were added.
35. Discussion, p12: Merge into the introduction, and add references to support “In turn, the middle layer is represented by two layers of collagen fibers, radially and circularly oriented.” this part was merged into the introduction,
ANSWER: The sentence was transferred; the references were added.
36. Discussion, p12: Merge into the introduction “These collagen fibers are responsible for the viscoelastic properties of the TM, which, in turn, characterize the TM mechanics [5,6].”
ANSWER: The sentence was transferred.
37. Discussion, p12: Merge into the introduction “The perforation of the TM leads to changes in its mechanical properties and hearing impairment. Therefore, in the field of otolaryngology, the search for materials with structures and mechanical properties similar to those of the TM, and their use as scaffolds to improve the regeneration of defects and restore the vibrational competence of the TM, remains a crucial issue.”
ANSWER: This part was transferred.
38. Discussion, p13: Merge into the introduction, and add references to support “Since 1990s, with the development of tissue engineering, synthetic materials have demonstrated a great potential in the TM restoration.”
ANSWER: The sentence was transferred; the references were added.
39. Discussion, p12: Merge into the introduction “The use of tissue engineered scaffolds provides significant advantages such as the reduction of the time of surgery and minimizing the surgical tissue damage, reduction of economic expenses, increase in the percentage of a positive morphofunctional result [5,7].”
ANSWER: The sentence was transferred.
40. Discussion, p13: Merge into the introduction, and add references to support “In spite of the TM’s inherent regenerative capacity, a support under the growing epithelial layer is required for its restoration.”
ANSWER: The sentence was transferred; the references were added.
41. Discussion, p13: Merge into the introduction, and add references to support “ Different scaffolds or transplants are applied in the TM bioengineering, which serve as supports to facilitate migration of cells and nutrients towards the perforation.”
ANSWER: The sentence was transferred; the references were added.
42. Discussion, p13: Merge into the introduction“ Opposite to conventional transplants retained in vivo, a bioengineered construct is meant to degrade with time and to be replaced with a new tissue [8]. We view the perfect purpose of tissue engineering techniques in the restoration of the three-layered TM structure which is crucial for sustaining its mechano-acoustic properties.”
ANSWER: This part was transferred.
43. Discussion, p13: Merge into the introduction, and add references to support “ Various scaffolds, stem cells and growth factors are used for the TM regenertion.”
ANSWER: The sentence was transferred; the references were added.
44. Discussion, p13: Merge into the introduction “The above materials have been applied both separately and in various combinations, according to the data of different studies [8–10].”
ANSWER: The sentence was transferred.
45. Discussion, p13: Merge into the “ All these components play a vital role both for cell differentiation, proliferation and migration and for the creation of a proper biochemical microenvironment to prepare and stimulate the tissue healing [5,11,12].”
ANSWER: The sentence was transferred.
46. Discussion, p13: Add references to support “By now a large number of scaffolds have been tested in the closure of both acute and chronic TM perforations”
ANSWER: The references were added.
47. Discussion, p14: It is unclear if Initial SBA-EPD collagen membrane could induce similar effects than the porous collagen application in “It is important that no notable scarring was observed at the sites of healed perforation in almost all the cases of porous collagen application, in both acute and chronic perforation groups. The material showed a good biodegradability with the preserved macrostructural integrity on day 14 and almost complete resorption on day 35.”
ANSWER: The sentences were corrected.
48. Discussion, p14: It is unclear how lack of inflammation was monitored to support “Of interest are the results of the histological study. First of all, no inflammatory infiltration was observed in all the samples after the collagen graft application. This finding indicates that the material was biologically compatible with the cellular and tissue structure of the TM and, moreover, prevented the addition of a bacterial infection of the middle ear. “
ANSWER: We corrected the sentences as follows: “The histological study of tissue reaction to the collagen graft did not reveal signs of the induced toxic effects or significant foreign body reaction. In particular, the tissues surrounding the implant were not infiltrated by leucocytes or macrophages in all the samples.”
49. Discussion, p14: Rephrase. There were only measurements on group 4 and it was not compared with other groups to adequately support “The mentioned data allow one to assume that, besides the potential repair of the fibrous layer, the TM’s vibrational properties also return to their normal state, however, additional studies are needed.”
ANSWER: The description was expanded. In group 3, there were no cases of the spontaneous closure of the TM perforation; therefore, it was not possible to measure the vibrational properties of the perforated TM. The indicated statement was removed.
50. Discussion, p14: Further experiments are needed to confirm “The above data make it possible to assume that it is the stimulation of differentiation and proliferation of epithelial progenitor cells in the TM that plays the main role in the closure of both acute and chronic perforations [46,47].”
ANSWER: The sentence was corrected.
51. Discussion, p14: Further experiments are needed to confirm “ “Thus, the presence of the foci of chondrogenesis may be, to some extent, related to the porous collagen graft’s function as a scaffold for progenitor cells of the TM forming a favorable matrix for their migration and proliferation not only to fibroblasts but also to chondroblasts.”
ANSWER: The statement was removed.
52. Discussion, p14: Add references to support “Accordingly, this finding is not negative when extrapolating the experimental conditions to clinical practice, since there is a long-standing practice of application of autologous perichondrium and cartilage implants in tympanoplasty for the defect closure and strengthening of the neo-TM.”
ANSWER: The references were added.
53. Discussion, p14: Add references to support “The eardrum plays an important role as the first and highly sensitive structure that perceives sound energy, transforming it into corresponding mechanical vibrations of further elements of the sound transmission chain.”
ANSWER: The references were added.
54. Discussion, p14: Add references to support “Pathological changes, persistent chronic perforation has a significant impact on the mechanical behavior of the TM.”
ANSWER: The references were added.
55. Discussion, p14-15: Apparently group4 was not compared with other groups. Rephrase “In this study, the highly sensitive vibrometer with a laser fiber-optic probe revealed decrease in high frequency threshold in the restored tympanic membranes up to 10 kHz comparing to the intact membranes”
ANSWER: The description was expanded. In group 3, there were no cases of the spontaneous closure of the TM perforation; therefore, it was not possible to measure the vibrational properties of the perforated TM. The indicated sentence was corrected as follows: “In this study, the highly sensitive vibrometer with a laser fiber-optic probe revealed a decrease in high frequency threshold in the restored TM with acute perforations up to 12kHz and chronic perforations up to 10 kHz comparing to the intact TM.”
Reviewer 2 Report
Comments and Suggestions for Authors
A collagen graft shaped as a sponge through SBA-EPD was used to treat acute and chronic TM perforations in a chinchilla model in this study. The authors said that the porous collagen scafold successfully enhanced TM regeneration, showing high biocompatibility and biodegradation potential. I think these issues need to be considered before publication.
1. There are some spelling errors in the article.
2. The pores within collagen matrix body were not interconnected. How does the author prove that the pores are not interconnected?
3. How did the authors treat the TM perforation with porous collagen membrane? How to make sure the collagen membrane covers the TM perforation and doesn't fall off?
4. Figure 4 showed that only group 3 and group 4 had significant differences in inflammation. So what are the advantages of the porous collagen membrane for TM perforations treatment?
5. The data of the TM’s vibrational properties and amplitude-frequency characteristics studies for Group 1 to Group 4 should be provided and then analyzed.
6. Authors should provide the ethical approval for animal experiments.
Author Response
Reviewer 2
A collagen graft shaped as a sponge through SBA-EPD was used to treat acute and chronic TM perforations in a chinchilla model in this study. The authors said that the porous collagen scafold successfully enhanced TM regeneration, showing high biocompatibility and biodegradation potential. I think these issues need to be considered before publication.
- There are some spelling errors in the article.
ANSWER: The paper was carefully proof read.
- The pores within collagen matrix body were not interconnected. How does the author prove that the pores are not interconnected?
ANSWER: We added a microphoto (Figure A3) showing a honeycomb-like structure of the collagen matrix at cross section. Under term “not interconnected”, we meant that the pores at cross section were distinguished (separate) and not directly connected to each other or to the superficial layers of the matrix.
- How did the authors treat the TM perforation with porous collagen membrane? How to make sure the collagen membrane covers the TM perforation and doesn't fall off?
ANSWER: Section 2.3.2 was expanded. On Day 14, we also performed endo-otoscopy, which showed the good membrane adhesion in all treated animals.
- Figure 4 showed that only group 3 and group 4 had significant differences in inflammation. So what are the advantages of the porous collagen membrane for TM perforations treatment?
ANSWER: We described both structural and functional benefits of using the collagen membrane. Its application led to the formation of the tympanic membrane resembling the normal structure in shape and microstructure.
Section 3.3: “However, there was a trend (p = 0.07) towards a decrease in TM thickness and an increase in collagen fibre density in group 4 compared to group 3.”
Section 4: “It should also be noted that the restored TM was presented mainly by dense regions of fibrous tissue consisting of tightly packed collagen fibers with a parallel orientation and a small number of fibroblasts between them. Thus, the structure of the restored TM fragment was close to that of an intact TM. The mentioned data allow one to assume that, besides the potential repair of the fibrous layer, the TM’s vibrational properties also return to their normal state, however, additional studies are needed.”
- The data of the TM’s vibrational properties and amplitude-frequency characteristics studies for Group 1 to Group 4 should be provided and then analyzed.
ANSWER: Sections 2.7 and 3.4 were extended. Data regarding group 1 and 2 were included. In group 3, there were no cases of the spontaneous closure of the TM perforation; therefore, it was not possible to measure the vibrational properties of the perforated TM.
Section 2.7: “Using this technique, we measured three intact TM and experimental ones: group 1 – acute perforation, control, n=3 (spontaneous closure in 35 days); group 2 – acute perforation, porous collagen membrane, n=3 (closure in 35 days); and group 4 – chronic perforation, porous collagen membrane, n=3 (closure in 35 days). In group 3, there were no cases of the spontaneous closure of the TM perforation; therefore, it was not possible to measure the vibrational properties of the perforated TM. The number of independent measurements was five”.
Section 3.4: “Using the experimental setup with a laser fiber optic probe, we studied the vibrational properties and AFC of the intact TM and TM with acute and chronic perforations closed spontaneously or due the treatment (group 1, 2, and 4). In group 3, there were no cases of the spontaneous closure of the TM perforation; therefore, it was not possible to measure the vibrational properties of the perforated TM…
In group 1, we revealed a drop in the sensitivity at middle and high frequencies (upper boundaries: 1, 2, 4 kHz) and a decrease in maximum response amplitudes by 30 dB, 25 dB, and 25 dB, respectively. The decrease in sensitivity at high frequencies and in maximum response amplitudes is typical for TM scars causing an increase in tissue stiffness. The vibrational TM characteristics in group 2 at the upper and lower limits of the frequency range were similar to those measured for the intact TM in two samples (1. ~ 100 Hz – 15000 Hz, 65 dB; 2. ~100 Hz-20000 Hz, 65 dB); one sample showed a slight decrease at the upper frequency limit (~100 Hz – 12000 Hz, 70 dB).
Analysis of the TM with the chronic perforation treated with the collagen matrix (group 4) showed that the upper and lower boundaries of the frequency audibility range appeared to be not lower than 10 kHz and not higher than 100 Hz, respectively. The highest amplitudes for all samples were 80 dB, 65 dB, and 70 dB, i.e. the deviation from the amplitude of the intact TM did not exceed 15 dB”.
- Authors should provide the ethical approval for animal experiments.
ANSWER: The animal experiments were approved by the local Ethical Committee at the Sechenov University (Protocol No. 11-23, June 15, 2023) (Section 2.3.1).
Reviewer 3 Report
Comments and Suggestions for Authors
The shrinkage temperature of both forms of collagen matrix was found be 54-56 °C. What should be the maximum limit of shrinkage temperature for repair of TM regeneration applications?
Conclusion, Para 599: The sentence “The TM regeneration… .. epithelial TM cells’ has to be reframed more meaningfully.
It is suggested to include future scope of the present work in conclusion section.
Author Response
Reviewer 3
The shrinkage temperature of both forms of collagen matrix was found be 54-56 °C. What should be the maximum limit of shrinkage temperature for repair of TM regeneration applications?
ANSWER: The shrinkage temperature indirectly reflects the degree of collagen crosslinking. In our case, the shrinkage temperature was ~ 54-56 °C which corresponds to a low crosslinking degree (10.1007/s10853-023-08641-x). Such matrices will biodegrade without causing the foreign body response that leads to pro-fibrotic complications. Rising the crosslinking degree (and shrinkage temperature, respectively) to values of around 65 °C will cause a pro-fibrotic response (10.1007/s10853-023-08641-x), which is not desired in the TM regeneration. We consider that collagen matrices with the shrinkage temperature not exceeding 60 °C can be used for the TM restoration.
Conclusion, Para 599: The sentence “The TM regeneration… .. epithelial TM cells’ has to be reframed more meaningfully.
ANSWER: The sentence was corrected.
It is suggested to include future scope of the present work in conclusion section.
ANSWER: The conclusion was expanded.
Round 2
Reviewer 1 Report
Comments and Suggestions for Authors
Reviewer’s answer: There are one major comment (I) and three minor comments (9, 12, 14) which remain insufficiently addressed.
General comments:
I. Porous collagen graft was tested to close acute and chronic tympanic membrane perforations in a chinchilla model. Otoscopic assessments were conducted on days 14 and 35, with histological evaluations and tympanic membrane vibrational determinations. The findings appear to be incomplete due to a lack of comparisons with other graft compounds as well as with SBA-EPD collagen membrane.
ANSWER: The matrix porosity is one of the main parameters ensuring better connective tissue ingrowth and vascularization and, therefore, quality of regeneration (10.1088/1758-5090/ac88a1, 10.1089/ten.tea.2010.0571). Thus, this parameter is considered to be obligatory that makes the developed matrix capable to be used to restore the TM perforations.
Reviewer’s answer: It was unaddressed. There were no comparisons with other graft compounds and with SBA-EPD collagen membrane.
Minor comments:
9. Materials and Methods, Creation of the collagen matrix, Preparation of a collagen suspension, p2: Indicate pH in “The tendon fragments underwent four 12-h- long treatments with 0.5 M NaCl, followed by homogenization in 0.8 M acetic acid.”
ANSWER: The requested information was included.
Reviewer’s answer: The pH of the suspension was 3.24, which may impact epidermal cell viability and function. This needs to be verified.
12. Materials and Methods, Creation of the collagen matrix, Preparation of a collagen suspension, p2-3: Indicate pH in both solutions. It was unclear why dialysis were performed at different pH in “The deposited collagen was redisolved in 0.8 M acetic acid and dialyzed against 0.5 M acetic acid for 3 days, the dialysis solution being replaced every day.”
ANSWER: The requested information was included. Initially lower pH (2.7) was required for better collagen dissolution, while the dialysis allowed for adjusting pH to optimal values for further SBA-EPD (pH 2.9).
Reviewer’s answer: It was unclear if distinct ionic strengths could impact the reproducibility of the dialysis. In addition, the acidic preparation may impact epidermal cell viability and function. This needs to be verified.
14. Materials and Methods, Collagen matrix characterization, p3: Phosphate tends to precipitate most polyvalent cations and is either a metabolite or an inhibitor in many systems (Good, et al., Hydrogen ion buffers for biological research. Biochemistry. 1966;5:467-477.). Justify the use of PBS buffer, indicate composition and pH to support “To measure the swelling values,the dried samples of collagen matrixwere first 1weighed, then they were placed in PBS at 4 °C for 8h.”
ANSWER: The required information was added. The PBS was purchased from Sigma Aldrich (Germany) and was ready to be used. The collagen in the membrane is in the “precipitated” state, so no changes will occur after its placing into PBS except hydration, which extent is measured in the “swelling” test. Testing collagen samples swelling in PBS is a standard procedure described elsewhere (10.1016/j.biomaterials.2003.09.066).
Reviewer’s answer: Ions could affect the swelling effects of collagen. Indeed non-collagenous matrix components are crucial to maintain collagen samples within a suitable and controlled environment. PBS may not correspond to the best buffer for the preparation of collagen. Other buffers types of buffers, shall be tested.
Author Response
Reviewer 1
There are one major comment (I) and three minor comments (9, 12, 14) which remain insufficiently addressed.
General comments:
- Porous collagen graft was tested to close acute and chronic tympanic membrane perforations in a chinchilla model. Otoscopic assessments were conducted on days 14 and 35, with histological evaluations and tympanic membrane vibrational determinations. The findings appear to be incomplete due to a lack of comparisons with other graft compounds as well as with SBA-EPD collagen membrane.
ANSWER: The matrix porosity is one of the main parameters ensuring better connective tissue ingrowth and vascularization and, therefore, quality of regeneration (10.1088/1758-5090/ac88a1, 10.1089/ten.tea.2010.0571). Thus, this parameter is considered to be obligatory that makes the developed matrix capable to be used to restore the TM perforations.
Reviewer’s answer: It was unaddressed. There were no comparisons with other graft compounds and with SBA-EPD collagen membrane.
ANSWER: The aim of our study was to develop a collagen-based material capable to induce the efficient morphological and functional regeneration in acute and chronic tympanic membrane (TM) perforations. Therefore, the assessment of influence of pores in a membrane was out of its scope as to restore the TM, the matrix has to be porous to ensure tissue ingrowth, so its porosity was considered as an obligatory requirement to be orthotopically implanted (10.1089/ten.TEA.2016.0265; 10.1089/ten.teb.2020.0176; 10.1016/j.pmatsci.2022.100942). This limitation was highlighted in the Discussion. The in vivo experiments described in our manuscript here was designed in accordance with the principles of the 3Rs (Replacement, Reduction, and Refinement).
Minor comments:
- Materials and Methods, Creation of the collagen matrix, Preparation of a collagen suspension, p2: Indicate pH in “The tendon fragments underwent four 12-h- long treatments with 0.5 M NaCl, followed by homogenization in 0.8 M acetic acid.”
ANSWER: The requested information was included.
Reviewer’s answer: The pH of the suspension was 3.24, which may impact epidermal cell viability and function. This needs to be verified.
ANSWER: Section 2.1.3 was expanded:
“…with 0.05% glutaraldehyde in PBS (Sigma-Aldrich, USA) for 30 min; then they were placed into fresh PBS and mechanically perforated on both sides using a cosmetic mesoroller with titanium needles, washed with PBS, and lyophilized at -40 °C for 24 hours (Figure A1C). As a result, the pH of their extracts was around 6.5-7.5…”
The biocompatibility of such membranes was described in our previous paper (10.1007/s10853-023-08641-x) and included both in vitro and in vivo experiments.
- Materials and Methods, Creation of the collagen matrix, Preparation of a collagen suspension, p2-3: Indicate pH in both solutions. It was unclear why dialysis were performed at different pH in “The deposited collagen was redisolved in 0.8 M acetic acid and dialyzed against 0.5 M acetic acid for 3 days, the dialysis solution being replaced every day.”
ANSWER: The requested information was included. Initially lower pH (2.7) was required for better collagen dissolution, while the dialysis allowed for adjusting pH to optimal values for further SBA-EPD (pH 2.9).
Reviewer’s answer: It was unclear if distinct ionic strengths could impact the reproducibility of the dialysis. In addition, the acidic preparation may impact epidermal cell viability and function. This needs to be verified.
ANSWER: Section 2.1.1 and 2.1.3 was expanded.
“After each dialysis procedure, we measured the pH of the resulting solution. The reproducibility of our results is guaranteed by the performed pH measurements, and after each dialysis, the pH of the solutions was stably equal to ~ 2.9. Therefore, we concluded that the performed dialysis procedure is well standardized…”
“…with 0.05% glutaraldehyde in PBS (Sigma-Aldrich, USA) for 30 min; then they were placed into fresh PBS and mechanically perforated on both sides using a cosmetic mesoroller with titanium needles, washed with PBS, and lyophilized at -40 °C for 24 hours (Figure A1C). As a result, the pH of their extracts was around 6.5-7.5…”
The biocompatibility of such membranes was described in our previous paper (10.1007/s10853-023-08641-x) and included both in vitro and in vivo experiments.
- Materials and Methods, Collagen matrix characterization, p3: Phosphate tends to precipitate most polyvalent cations and is either a metabolite or an inhibitor in many systems (Good, et al., Hydrogen ion buffers for biological research. Biochemistry. 1966;5:467-477.). Justify the use of PBS buffer, indicate composition and pH to support “To measure the swelling values,the dried samples of collagen matrix were first 1weighed, then they were placed in PBS at 4 °C for 8h.”
ANSWER: The required information was added. The PBS was purchased from Sigma Aldrich (Germany) and was ready to be used. The collagen in the membrane is in the “precipitated” state, so no changes will occur after its placing into PBS except hydration, which extent is measured in the “swelling” test. Testing collagen samples swelling in PBS is a standard procedure described elsewhere (10.1016/j.biomaterials.2003.09.066).
Reviewer’s answer: Ions could affect the swelling effects of collagen. Indeed non-collagenous matrix components are crucial to maintain collagen samples within a suitable and controlled environment. PBS may not correspond to the best buffer for the preparation of collagen. Other buffers types of buffers, shall be tested.
ANSWER: Section 2.2 was expanded:
“…The swelling test was performed in accordance with the protocols described elsewhere [35–37] …”
PBS was not used to prepare collagen suspension or collagen membranes.
Reviewer 2 Report
Comments and Suggestions for Authors
Accept
Author Response
Reviewer 2
Accept
ANSWER: We would like to express their gratitude to the reviewer for the time spent and valuable comments that surely allowed us to improve the quality of the paper.
Round 3
Reviewer 1 Report
Comments and Suggestions for Authors
The concerns were addressed.